# End-To-End Causal Effect Estimation from Unstructured Natural Language Data

**Nikita Dhawan**[*]
University of Toronto, Vector Institute
nikita@cs.toronto.edu

**Leonardo Cotta**
Vector Institute
leonardo.cotta@vectorinstitute.ai

**Karen Ullrich**
Meta AI
karenu@meta.com

**Rahul G. Krishnan**
University of Toronto, Vector Institute
rahulgk@cs.toronto.edu

**Chris J. Maddison**
University of Toronto, Vector Institute
cmaddis@cs.toronto.edu

## Abstract

Knowing the effect of an intervention is critical for human decision-making, but current approaches for causal effect estimation rely on manual data collection and structuring, regardless of the causal assumptions. This increases both the cost and time-to-completion for studies. We show how large, diverse observational text data can be mined with large language models (LLMs) to produce inexpensive causal effect estimates under appropriate causal assumptions. We introduce *NATURAL*, a novel family of causal effect estimators built with LLMs that operate over datasets of unstructured text. Our estimators use LLM conditional distributions (over variables of interest, given the text data) to assist in the computation of classical estimators of causal effect. We overcome a number of technical challenges to realize this idea, such as automating data curation and using LLMs to impute missing information. We prepare six (two semi-synthetic and four real) observational datasets, paired with corresponding ground truth in the form of randomized trials, which we used to systematically evaluate each step of our pipeline. NATURAL estimators demonstrate remarkable performance, yielding causal effect estimates that fall within 3 percentage points of their ground truth counterparts, including on real-world Phase 3/4 clinical trials. Our results suggest that unstructured text data is a rich source of causal effect information, and NATURAL is a first step towards an automated pipeline to tap this resource.

## 1   Introduction

Estimating the causal effects of interventions is time consuming and costly, but the resulting outcomes are precious. Health agencies around the world often require randomized controlled trial (RCT) data to approve medical interventions. Clinical trials are key contributors to large R&D costs for drug developers [30]. Natural experiments are another source of rich interventional data, but they may not always exist or have enough data relevant to a given causal hypothesis [12].

When treatment randomization is infeasible, observational data can be used to identify average treatment effects (ATEs) [48], under common assumptions, *e.g.*, no unobserved confounding. Such

---

[*]Work done during internship at Meta AI.

38th Conference on Neural Information Processing Systems (NeurIPS 2024).

| Financial cost | Study cohort(s) | Treatment assignment | Observed data | Example estimator & causal graph |
|---|---|---|---|---|
| **Completely randomized experiment** $$$ | | | (T Y) | Difference-in-Means |
| **Observational structured experiment** $$ | | | (T Y X1 X2 X3) | IPW |
| **Observational natural language experiment** $ | | | I tried a drug and it helped! I haven't had a single … | **NATURAL** |

Figure 1: When compared to experimental and other observational studies, NATURAL has lower costs and provides greater diversity in cohort selection, for causal effect estimation.

data is abundant but even when the necessary assumptions are satisfied, it must be *structured* (*i.e.*, the outcomes, treatments, and relevant covariates must be defined, recorded, and tabulated) before it becomes amenable to computational analyses.

Yet, unstructured observational data presents unique opportunities for cheaper, more accessible, and potentially even better [32] effect estimation. For example, thousands of people living with diabetes choose to share their experiences with *treatments* on online patient forums. Some of their posts contain rich descriptions of daily lives, the drugs they have been prescribed, the treatment responses and side effects, as well as pre-treatment information like age and sex. Their posts contain their lived experiences including evidence of an *outcome* in an observational experiment, albeit in an unstructured form. Other potential sources of rich unstructured, observational data include newspaper classifieds, police reports, social media, and clinical reports. Despite being collected for a myriad of purposes, researchers have often turned to such data to test hypotheses since: (i) unstructured data does not require restrictive data collection designs, *e.g.*, measurement choice, and can admit many different post-hoc analyses; (ii) the reported outcomes may reflect what matters to subjects better than standard outcome measures; (iii) value may be recovered from outcomes that would otherwise be lost; (iv) there may be *more* unstructured data available on underserved or marginalized populations. Figure 1 contrasts our setting with previous works using randomized or structured observational data.

This work asks a simple question: *How can we use large language models to automate treatment effect estimation using freely available text data?* We introduce *NATURAL*, a family of text-conditioned estimators that addresses this by performing *NATural language analysis to Understand ReAL effects*.

At a high level, the steps required to compute NATURAL estimators are as follows. Given an observational study design and a dataset of natural language reports, filter for reports that are likely to conform to the experimental design. Then, using a large language model (LLM), extract the conditional distribution of structured variables of interest (outcome, treatment, covariates) given the report. Finally, use the conditionals to compute estimators of the ATE, using classical strategies such as inverse propensity score weighting and outcome imputation.

NATURAL is a data-driven pipeline. It leverages and relies on the LLM in a manner that mimics the learning task it was trained for: providing parametric approximations to conditional distributions. As in all observational studies, the validity of NATURAL also depends on prior causal knowledge about the task. Expert knowledge is required to define appropriate covariates and confirm that they satisfy the necessary assumptions for effect estimation. However, we anticipate that NATURAL estimators could be developed under other structural assumptions (e.g. instrumental variables) as well.

The core contributions of our work are:

- We derived NATURAL ATE estimators based on classical estimators of the ATE, like inverse propensity score weighting and outcome imputation. NATURAL estimators operate on entirely unstructured data under two novel data-access assumptions.
- We implemented NATURAL estimators using an LLM-based pipeline.
- We developed six observational datasets to systematically evaluate parts of this pipeline: two semi-synthetic datasets constructed using marketing data, and four clinical datasets curated from public (pre-December 2022) migraine and diabetes subreddits from the Pushshift collection [7].
- For each dataset, we treated the ATE from a corresponding real-world completely randomized experiment (CRE) as ground truth. Remarkably, our predicted ATEs all fell within 3 percentage points of the ground truth ATEs, a potential cost savings of many millions of dollars.

## 1.1 Related Work

Leveraging natural language data [42] to support causal claims is pervasive in applied research [43, 13]. Our work falls under the broad umbrella of accelerating the identification of real-world evidence (RWE) [40]. For instance, in the context of healthcare, RWE supports not only drug repurposing, but also post-market safety evaluations — its most common application. NATURAL expands the boundaries of how quickly one can obtain and validate such real-world evidence from observational data [41].

The use of natural language data in causal inference comes in different flavors: i) using text to measure confounders [23], ii) using text to measure causal effect outcomes [15], or iii) producing interpretable causal features from text [15, 6], *e.g.*, what words are more likely to explain the cause of an event. NATURAL distinguishes itself from these lines of research in two ways: i) NATURAL does not require any curated task-specific training data (it is zero-shot), and ii) NATURAL is not interested in how the text itself, *i.e.*, its words, relate to the causal problem —that is, we are only leveraging the model's ability to predict the distribution of a specified variable conditional on the input text. We highlight that our work lies distinct from research at the intersection of text and causality that combines text and numerical or tabular data [14], the latter of which may be unavailable or incomplete in settings involving neglected diseases, unrecorded abortions, or illicit drug use. Other work has also studied topic modelling approaches [2] or the ability of language models to infer *latent* variables (that are implied but not explicitly identified in text data) [36, 13]. Rather, we require the precise specification of covariates to condition on – we view this as being crucial to creating a more direct way for an end user to verify the validity of information extracted with our approach.

Prior works have also leveraged LLMs in a black-box fashion for causal tasks by querying the model for causal statements. In the context of causal discovery, users directly ask for the existence of cause-and-effect relationships, *e.g.*,"Does changing the age of an abalone causes a change in its length?" [25, 33, 4, 5, 44, 21, 6]. Due to the large amount of training data, it is possible that the model learns to apply a causal model described in the training data and answer causal questions with it [35, 47]. The issue with this approach is i) the user is limited to the causal models observed in training, ii) the user is not aware of *which* causal model they are using, and iii) the queries tend to present high prompt sensitivity [28]. Finally, we note that a recent work created a benchmark and showed how LLMs struggle to distinguish pairwise correlation from causation [20], while another shows that checking causal relationships in a pairwise manner can lead to invalid causal graphs [45].

## 2 Preliminaries

We are interested in estimating the causal effect of a treatment relative to either another treatment or no treatment in a population of interest. More precisely, we consider treatments $t \in \{0, 1\}$ and the corresponding potential outcomes $Y(1)$ and $Y(0)$ under each treatment. We wish to compute the quantity $\tau := \mathbb{E}[Y(1) - Y(0)]$, often referred to as Average Treatment Effect (ATE). Sometimes, $Y(0)$ may correspond to no treatment (control). Throughout this work, we assume binary treatments and outcomes in the Neyman-Rubin causal model. We provide a full list of notation in appendix A.

A Completely Randomized Experiment (CRE) with $n$ participants requires no prior causal knowledge. In a CRE, the treatment assignment vector $(\tilde{T}_i)_{i=1}^n$ is a random permutation of $n_1$ ones and $n - n_1$ zeros sampled independently of the outcomes. In this case, the difference-in-means $\frac{1}{n_1}\sum_{i=1}^n \tilde{T}_i Y_i(1) - \frac{1}{n-n_1}\sum_{i=1}^n (1 - \tilde{T}_i)Y_i(0)$ provides us with an unbiased estimate of $\tau$.

Despite the indisputable necessity of CREs in high-stakes settings, it is often expensive and/or infeasible to have complete control over the treatment assignment. Instead, *observational* data is more readily available. Observational data often contains spurious correlations between the observed treatment $T$ and the observed outcome $Y = TY(1) + (1 - T)Y(0)$ through a common cause (confounder). Typically, this confounding is formalized as a variable $X$, which we assume to be discrete throughout this work, representing covariates associated with each individual. Given i.i.d. samples $\{(X_i, T_i, Y_i)\}_{i=1}^n$ from the target population, standard causal inference techniques can correct for confounding bias and provide consistent estimates of $\tau$ under Assumptions 1 and 2:

**Assumption 1 (Strong Ignorability.)** *The potential outcomes are independent of treatment assignments conditional on covariates, i.e., $(Y(0), Y(1)) \perp\!\!\!\perp T|X$.*

**Assumption 2 (Positivity.)** *For every treatment $t$ and covariate set $x$, $0 < P(T = t \mid X = x) < 1$.*

Following are two classical estimators of the ATE $\tau$ from observational data, each of which rely on $X$ satisfying Assumptions 1 and 2. We refer the reader to Ding [11] for further details.

**Inverse Propensity Score Weighting (IPW).** The propensity score is the conditional probability of receiving a treatment given the observed features, *i.e.*, $e(x) = P(T = 1|X = x)$. The IPW estimator is given by

$$\hat{\tau}_{\text{IPW}} = \frac{1}{n} \sum_{i=1}^{n} \frac{T_i Y_i}{\hat{e}(X_i)} - \frac{(1 - T_i)Y_i}{1 - \hat{e}(X_i)}, \tag{1}$$

where $\hat{e}(x)$ is an approximation of $P(T = 1 \mid X = x)$. When $\hat{e}(x)$ is the true propensity score, $\hat{\tau}_{\text{IPW}}$ is an unbiased estimator of $\tau$. When $\hat{e}(x)$ is estimated as empirical probability, $\hat{\tau}_{\text{IPW}}$ is consistent.

**Outcome Imputation (OI).** Outcome Imputation learns a model to impute outcomes from features and treatment and then marginalizes away the features to estimate $\tau$ with

$$\hat{\tau}_{\text{OI}} = \frac{1}{n} \sum_{i=1}^{n} \hat{\tau}(X_i, 1) - \hat{\tau}(X_i, 0), \tag{2}$$

where $\hat{\tau}(x, t)$ approximates $P(Y = 1 \mid X = x, T = t)$. Note that if $\hat{\tau}(x, t)$ is an unbiased estimation of this quantity, $\hat{\tau}_{\text{OI}}$ is an unbiased estimator of $\tau$.

# 3 NATURAL estimators of the ATE

Both CRE and observational studies require direct access to tabulated data $(X_i, T_i, Y_i)$ for every individual $i$. Our NATURAL estimators on the other hand estimate the ATE from observational, unstructured natural language data in the form of language reports $R_i$. In addition to Assumptions 1 and 2, NATURAL estimators require the following assumptions to guarantee their consistency.

**Assumption 3 (Natural language report data.)** *The target population is described by an observational data-generating process $P(X, T, Y, R)$ of data $(X, T, Y)$, which satisfies Assumptions 1 and 2 and is jointly distributed with a random natural language string $R$, called a* report. *We assume access to an i.i.d. sample of reports $\{R_i\}_{i=1}^{n}$ from the marginal of this process.*

**Assumption 4 (Access to the true observational conditional over $(X, T, Y)$.)** *We can either (i) compute the conditional $P(X = x, T = t, Y = y|R = r)$ of the true data-generating process, or (ii) we can sample from $P(X = x|R = r)$ and compute $P(T = t, Y = y|R = r, X = x)$.*

Intuitively, these assumptions give NATURAL indirect access to $(X, T, Y)$ through $R$. They can be weak or strong, depending on the definition of the reports $R$. On the one hand, if reports are copies of the observational data, *i.e.*, $R = (X, T, Y)$, then Assumption 4 is trivial to satisfy. On the other hand, if reports are all the constant, empty string, $R = \epsilon$, then Assumption 4 guarantees that we have full access to the *true* observational joint density function over $(X, T, Y)$, which is a strong assumption. In other words, it requires a way to simulate trial outcomes unconditionally (without any data). We consider how we might satisfy these assumptions in practice in the next section. Here, we assume that they hold and develop a series of consistent estimators of the ATE.

**NATURAL Full.** Given $\{R_i\}_{i=1}^{n}$ and $P(X = x, T = t, Y = y|R = r)$, we can construct an idealized version of NATURAL. Let us start by noting that the law of total expectation gives us

$$\tau = \mathbb{E}_{X,T,Y} \left[ \frac{TY}{e(X)} - \frac{(1 - T)Y}{1 - e(X)} \right] = \mathbb{E}_R \left[ \mathbb{E}_{X,T,Y|R} \left[ \frac{TY}{e(X)} - \frac{(1 - T)Y}{1 - e(X)} \right] \right]. \tag{3}$$

A Monte Carlo estimate over reports is given by

$$\hat{\tau}_{\text{N-Full}} = \frac{1}{n} \sum_{i=1}^{n} \sum_{x,t,y} P(X = x, T = t, Y = y|R_i) \left[ \frac{ty}{\hat{e}_{\text{N-Full}}(x)} - \frac{(1 - t)y}{1 - \hat{e}_{\text{N-Full}}(x)} \right], \tag{4}$$

which further approximates $\hat{e}_{\text{N-Full}}(x)$ from the given conditional. We used eq. (7) below. We note that $\hat{\tau}_{\text{N-Full}}$ above is derived from IPW, but can also be derived from OI, as shown in appendix B.

The estimator $\hat{\tau}_{\text{N-Full}}$ above relies on enumerating all possible values of $(X, T, Y)$, making it computationally expensive for high-dimensional $X$. Below, we present two hybrid versions of our method which combine sampling of some variables and computation of conditional probabilities of others.

**NATURAL IPW.** To construct our hybrid estimator, we augment the data $\{R_i\}_{i=1}^n$ by sampling from $P(X|R_i)$ independently for each report $R_i$. This gives us a dataset $\{(R_i, X_i)\}_{i=1}^n$ drawn i.i.d. from $P(X, R)$ by Assumption 4. Then, our hybrid estimator is derived from the form of IPW as follows:

$$\tau = \mathbb{E}_{R,X}\left[\mathbb{E}_{T,Y|R,X}\left[\frac{TY}{e(X)} - \frac{(1-T)Y}{1-e(X)}\right]\right], \tag{5}$$

$$\hat{\tau}_{\text{N-IPW}} = \frac{1}{n}\sum_{i=1}^n \sum_{(t,y)\in\mathcal{T}\times\mathcal{Y}} P(T=t, Y=y|R_i, X_i)\left[\frac{ty}{\hat{e}_{\text{N-IPW}}(X_i)} - \frac{(1-t)y}{1-\hat{e}_{\text{N-IPW}}(X_i)}\right], \tag{6}$$

where $\hat{e}_{\text{N-IPW}}(x)$ is consistently estimated in the following manner:

$$\hat{e}_{\text{N-IPW}}(x) = \frac{\sum_{i=1}^n P(T=1|R_i, X_i)\mathbb{I}(X_i=x)}{\sum_{i=1}^n \mathbb{I}(X_i=x)} \xrightarrow{\text{a.s.}} \frac{\mathbb{E}_{R,X}\left[P(T=1|R,X)\mathbb{I}(X=x)\right]}{\mathbb{E}_{R,X}\left[\mathbb{I}(X=x)\right]} = e(x). \tag{7}$$

**NATURAL OI.** Similarly inspired by the OI estimator in equation 2, we have for $t \in \{0, 1\}$,

$$P(Y=1 \mid T=t, X=x) = \frac{\mathbb{E}_{R,X,T}\left[P(Y=1|R,X,T)\mathbb{I}(X=x, T=t)\right]}{\mathbb{E}_{R,X,T}\left[\mathbb{I}(X=x, T=t)\right]}. \tag{8}$$

Thus, for our hybrid OI estimator, we augment the data $\{R_i\}_{i=1}^n$ by sampling from $P(X, T|R_i)$ independently for each report $R_i$. This gives us a dataset $\{(R_i, X_i, T_i)\}_{i=1}^n$ drawn i.i.d. from $P(R, X, T)$ by Assumption 4. Then, our consistent outcome predictor is given by

$$\hat{\tau}_{\text{N-OI}}(x, t) = \frac{\sum_{i=1}^n P(Y=1|R_i, X_i, T_i)\mathbb{I}(X_i=x, T_i=t)}{\sum_{i=1}^n \mathbb{I}(X_i=x, T_i=t)}, \tag{9}$$

and the final estimator is given by:

$$\hat{\tau}_{\text{N-OI}} = \frac{1}{n}\sum_{i=1}^n \hat{\tau}_{\text{N-OI}}(X_i, 1) - \hat{\tau}_{\text{N-OI}}(X_i, 0) \tag{10}$$

**NATURAL Monte Carlo.** Further in the direction of sampling more variables, we can obtain samples $(X_i, T_i, Y_i)$ from $P(X, T, Y|R_i)$ and compute a Monte Carlo estimate, $\hat{\tau}_{\text{N-MC}}$. The set of samples $\{(X_i, T_i, Y_i)\}_{i=1}^n$ constitute a tabular dataset which can be plugged into a standard ATE estimator like IPW or OI, as described in section 2. We refer to these sample-only estimators as N-MC IPW and N-MC OI, respectively.

**Inclusion Criteria conditioned ATE.** We are sometimes interested in ATEs over populations defined by constraints on pre-treatment covariates $X_i$, known as *inclusion criteria* and denoted by $I$. This conditional ATE is $\tau(I) = \mathbb{E}[Y(1) - Y(0) \mid X \in I]$ and satisfies the following identity.

$$\tau(I) = \mathbb{E}_R\left[\frac{P(X \in I|R)}{P(X \in I)}\mathbb{E}_{X,T,Y}\left[\frac{TY}{e(X)} - \frac{(1-T)Y}{1-e(X)}\Bigg| X \in I, R\right]\right]. \tag{11}$$

The conditional version of NATURAL IPW can be estimated by filtering out reports where $P(X \in I|R_i) = 0$, sampling $X_{ij} \sim P(X|R_i, X \in I)$ i.i.d., and finally weighting datapoints by the relative likelihood of matching the inclusion criteria given the report:

$$\hat{\tau}(I) = \sum_{i=1}^n\left[\frac{P(X \in I|R_i)}{\sum_{i=1}^n P(X \in I|R_i)}\sum_{j=1}^m \frac{1}{m}\mathbb{E}_{T,Y|X_{ij}, R_i}\left[\frac{TY}{e(X_{ij})} - \frac{(1-T)Y}{1-e(X_{ij})}\right]\right], \tag{12}$$

where the inner expectation is estimated similar to eq. (6). A complete derivation for eqs. (11) and (12) and related discussion are included in appendix G. In practice, we took $m=1$.

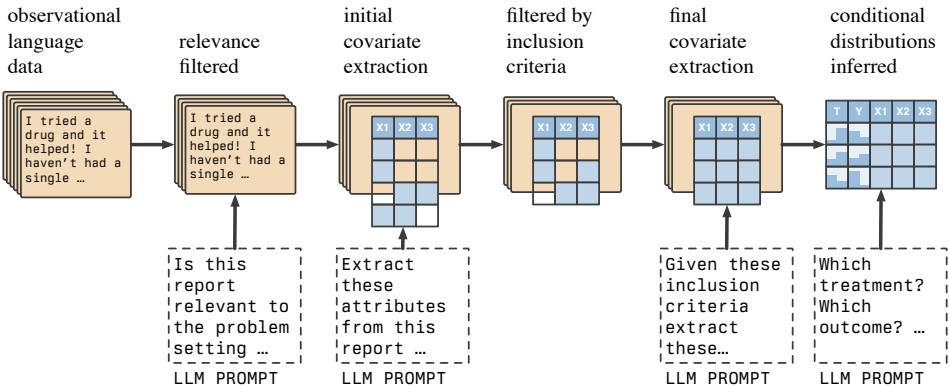

Figure 2: Our pipeline leverages LLMs to curate data that can be plugged into natural language conditioned estimators for average treatment effects.

## 4 Implementing NATURAL estimators with Large Language Models

LLMs are trained on vast datasets of real-world data, *e.g.*, [3], which likely contain records of data generated by processes that are consistent with Assumption 3. Because LLMs can learn well-calibrated conditionals [22], our hypothesis is that LLMs can be prompted to approximate the conditionals required by Assumption 4 for real-world causal effect questions of interest. Our LLM implementation of NATURAL estimators is built on this hypothesis to try to satisfy Assumptions 3 and 4 (Assumptions 1 and 2 must be guaranteed by a domain expert). We defer exact prompts for LLM inference to appendix D, a full worked example to appendix C, and a discussion of the limitations of our approach to the next section. Figure 2 summarizes our pipeline.

**Filtering to match Assumption 3.** For our real-data clinical settings, our first goal is to produce a dataset of i.i.d. reports $R_i$ that are very likely to be jointly distributed with the random variables $(X_i, T_i, Y_i)$ of a specific observational study of interest. Given a study of interest and a dataset of real-world reports that are potentially relevant to the study, we pass it through a sequence of filters with increasing detail and strictness:

 (i) **Initial filter.** Inspired by other work with social media data [1, 38], we first use deterministic rules to filter out uninformative reports: posts that were removed, are too short, have "bot" in the author's name, have no mention of any keyword related to the study, etc.
 (ii) **Filter by relevance.** We prompt an LLM to determine whether each report contains information that would make it relevant to the study. We remove reports that are deemed irrelevant.
 (iii) **Filter by treatment-outcome.** We ensure that each report pertains specifically to the treatments and outcomes of interest. We do so by prompting an LLM to extract only treatment and outcome information, and retaining only the posts that are deemed to both mention one of the treatments in question and also contain outcome information.
 (iv) **Filter known covariates by inclusion criteria.** For ATEs conditioned on inclusion criteria, as in our real-world datasets, we included a filter to enforce these criteria. Managing inclusion criteria is complicated by the fact that many reports $R_i$ contain no or partial information about covariates that are required to verify inclusion. So, in this filtering step, our goal was to ensure that the final set of reports have non-zero probability of matching the inclusion criteria. We begin by prompting an LLM to extract the full set of covariates $X_i$, following constraints on the possible values each attribute can take, but we allow the LLM to extract Unknown if it is impossible for the LLM to determine the value of a covariate. We then remove reports, if any of the non-Unknown covariates are determined to fail their inclusion criteria. We found the JSON-mode made available for generation by certain LLM APIs, to suffice for this task; however more involved strategies for constrained generation are also possible [46, 50].

**Sampling from and computing conditional probabilities to match Assumption 4.** Given a set of reports $\{R_i\}_{i=1}^n$ that pass the filtering stage above, our next steps use LLMs to extract the samples and conditionals $P_{\text{LLM}}(X, T, Y \mid R)$, required to compute NATURAL estimators. For each $R_i$, we:

 (v) **Extract covariates, both known and unknown.** We run a final covariate extraction by prompting an LLM to determine the full set of covariates $X_i$ from the report $R_i$, subject to the constraint that $X_i$ satisfies the inclusion criteria. In contrast to (iv), we ask the LLM to

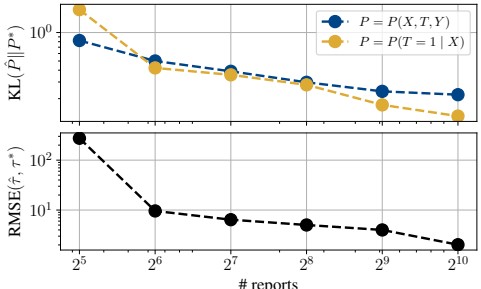 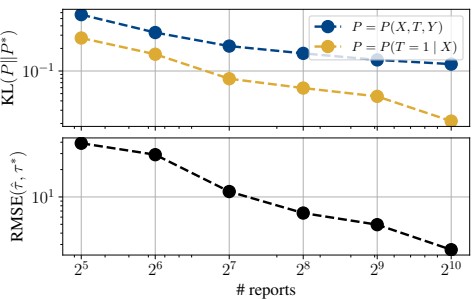

Figure 3: For Hillstrom (left) and Retail Hero (right), the KL divergence between estimated joint and propensity distributions and their true counterparts reduces with increasing number of posts (top), as does the RMSE between the NATURAL Full estimate and true ATE (bottom).

guess the values of Unknown covariates. We verified that this second extraction agreed exactly with the first extraction (iv) on the known covariates (*i.e.*, the ones that were not extracted as Unknown in the first extraction). We contrast the empirical distributions of these known and unknown/guessed covariates for our experiments in appendix F.1.

(vi) **Infer conditionals.** Given $\{R_i, X_i\}_{i=1}^n$ from the previous steps, we compute the probabilities $P_{\text{LLM}}(T = t, Y = y | R_i, X_i)$ by prompting an LLM that makes log-probabilities accessible. Specifically, we ask an LLM to answer questions about $T, Y$ given access to $R_i, X_i$, and we score every possible answer $T = t, Y = y$ using the LLM log-probabilities. We exponentiate and renormalize these scores across the space of possible realizations to obtain a valid probability distribution.

(vii) **Weight reports according to inclusion criteria match.** Similar to item (vi), we compute $P_{\text{LLM}}(X_i \in I | R_i)$ to obtain the weights in eq. (12), by prompting an LLM with descriptions of the inclusion criteria that must be satisfied and each report $R_i$. It may be possible to skip this weighting step under additional structural assumptions on the data. These assumptions as well as experimental results without the weighting are included in appendix G.

Nevertheless, while our empirical results are remarkably consistent with the correctness of our pipeline, we cannot formally guarantee that it satisfies Assumptions 3 and 4. The final outcome of this pipeline is a dataset $\{R_i, X_i\}_{i=1}^n$ and a set of conditionals $P(T = t, Y = y | R_i, X_i)$ that can be plugged into the hybrid NATURAL estimators in section 3 to predict ATEs. Therefore, we see this as a first implementation of NATURAL estimators, which we anticipate can be improved.

## 5 Limitations and Broader Impact

In addition to the limitations that NATURAL shares with every observational study, *i.e.*, the validity of the practitioner's causal assumptions, it comes with an *extra dependence on how well one can approximate the desired conditional distributions*. While more and more capable LLMs are being continually developed, the extent to which they satisfy NATURAL's assumptions is nearly impossible to formally test. Indeed, while pretraining tends to produce calibrated LLM predictions [22], post-training techniques can compromise calibration [34]. Therefore, we emphasize that NATURAL was *not* developed to recommend therapeutics directly to end-users or to directly inform high-stakes public policies. Instead, we envision NATURAL as a powerful tool to help us approximate ATEs at scale and prioritize confirmatory CREs. We strongly recommend that all predictions made by NATURAL estimators be validated experimentally before being used to inform high-stakes decision-making. Apart from its dependence on LLM capabilities, NATURAL is also limited by the nature of observational, unstructured natural language data:

- *Network Interference.* In practice, acquiring i.i.d. reports can be challenging. For instance, social network users might talk to each other and influence their treatment choices. This is a well-known issue in causal inference and statistical sciences in general. Existing solutions rely on a known network structure to sample individuals or correct for their neighbors' treatments [10, 26, 17].

- *Outcome Measurement.* Since NATURAL deals with self-reports, subjects need to be able to report the outcomes of interest. For example, this cannot be applied if the outcome is measured with an expensive, inaccessible test. Therefore, the study design implemented with NATURAL must account for the accessibility of endpoints to users.

- *Selection Bias.* Results might be biased towards individuals' choice of reporting an outcome given their experience with the treatment. Luckily, outcome missingness is a widely studied problem in causality research, see *e.g.*, how to test [9] or how to mitigate [31] it. Note, however, that solutions will often accumulate assumptions on top of NATURAL and should always be critically evaluated by practitioners. Finally, apart from individuals' choice of reporting, selection bias might also arise from which individuals participate in online forums, *i.e.*, our framework is only capable of estimating *local* ATEs —external validity is not guaranteed a priori. We demonstrate this challenge in section 6 by simulating systematic bias in semi-synthetic settings.

## 6  Empirical Evaluation

Evaluating an end-to-end pipeline for causal inference from unstructured real-world text data to ATEs presents challenges regarding access to data, ground truth ATEs and insightful intermediate metrics. We used two semi-synthetic datasets where we augmented randomized data to mimic real-world observations, while continuing to have access to ground truth evaluation. In addition, we study four real datasets, curated from publicly available Reddit posts from the PushShift dataset, as described in section 4. These six datasets allowed us to systematically evaluate NATURAL.

**Semi-synthetic Datasets.** Causal effect estimation is typically evaluated using synthetic datasets with one or more relationships between the observed covariates, treatment and outcome being contrived. We instead synthesized unstructured observational text data from real randomized tabular datasets, using an LLM. Specifically, we (i) introduced confounding bias by sampling datapoints according to an artificial propensity score, (ii) randomly dropped covariates, (iii) described covariates, treatment and outcome in shuffled orderings, (iv) simu-

Table 1: The NATURAL IPW ATE outperforms other versions of the method as well as trained baselines on semi-synthetic datasets, as measured by RMSE.

| | Hillstrom | | Retail Hero | |
|---|---|---|---|---|
| | ATE (%) | RMSE | ATE (%) | RMSE |
| **Selection-biased N-IPW** | $-3.49 \pm 1.46$ | 9.58 | $10.66 \pm 2.24$ | 7.67 |
| **Uncorrected** | $1.86 \pm 0.67$ | 4.28 | $0.26 \pm 0.30$ | 3.08 |
| **N-Full** | $4.26 \pm 0.86$ | 2.02 | $1.86 \pm 1.38$ | 2.08 |
| **N-MC OI** | $6.17 \pm 1.61$ | 1.61 | $4.94 \pm 2.17$ | 2.70 |
| **N-MC IPW** | $4.81 \pm 0.80$ | 1.51 | $1.85 \pm 2.01$ | 2.49 |
| **N-OI** | $4.58 \pm 0.61$ | 1.62 | $2.99 \pm 1.43$ | 1.72 |
| **N-IPW** | $\mathbf{5.23 \pm 1.00}$ | **1.32** | $\mathbf{3.83 \pm 1.29}$ | **1.39** |
| **Bag-of-Words** | $7.57 \pm 1.37$ | 2.23 | $2.61 \pm 2.08$ | 2.42 |
| **Sentence Encoder** | $0.00 \pm 0.00$ | 6.09 | $1.97 \pm 1.62$ | 2.10 |
| **IPW (Structured)** | $6.38 \pm 0.26$ | 0.39 | $3.09 \pm 0.19$ | 0.30 |
| **Ground Truth** | **6.09** [19] | - | **3.32** [49] | - |

lated realism by sampling a persona from the the Big Five personality traits [27] for each datapoint and finally, (v) prompted the LLM to generate a realistic report describing the provided information in the style of someone with the given traits (see appendix D for the full prompt). We used two standard, publicly available randomized datasets: **Hillstrom** [19] and **Retail Hero** [49], and plan to open-source scripts to generate our data. Step (i) above is in a similar vein as Keith et al. [24], in that our subsampling strategy does not modify the marginal distribution over covariates and the ATE remains identifiable from observational data.

**Real-world Datasets.** To study how our framework may be deployed to test hypotheses using real data from online forums; we considered two medical conditions for which there exist abundant Reddit posts in the PushShift collection [7], with individuals' personal experiences: the effect of diabetes medications (e.g. Semaglutide) on weight loss and the tolerability of migraine treatments. For each condition, we picked two clinical trials which performed head-to-head comparisons of two treatments that we expected to find references to in relevant subreddits. Moreover, to mitigate selection bias we selected pairs of similar treatments, *e.g.*, comparable availability, where we believe the probability of a user reporting their experience is approximately equal in both. As we will discuss, our results suggest external validity as well, meaning that the probability of a user reporting their experience with the treatments seems to be (approximately) equal to the prior probability of a user reporting any experience. We limited our data collection to posts that were written before December 2022 and made publicly available in the PushShift archives. We curated four datasets for comparison between different treatments, each of which has a ground truth RCT: **Semaglutide vs. Tirzepatide** [18] and **Semaglutide vs. Liraglutide** [8] for their effect on weight loss and **Erenumab vs. Topiramate** [37] and **OnabotulinumtoxinA vs. Topiramate** [39] for their tolerability. We used the first of these to validate implementation choices NATURAL (like filtering, imputations, prompt specifications) and

Table 2: Using real data, best performing NATURAL estimators fall within 3 percentage points of their corresponding ground truth clinical trial ATEs. Possible ATE values lie between $-100$ and $100$.

| | Tuned | | | | Held-out | | | |
|---|---|---|---|---|---|---|---|---|
| | Semaglutide vs. Tirzepatide (% weight loss $\geq$ 5%) | | Semaglutide vs. Liraglutide (% weight loss $\geq$ 10%) | | Erenumab vs. Topiramate (% discontinued due to AE) | | OnabotulinumtoxinA vs. Topiramate (% discontinued due to AE) | |
| | ATE (%) | RMSE | ATE (%) | RMSE | ATE (%) | RMSE | ATE (%) | RMSE |
| Uncorrected | $-33.56 \pm 0.77$ | 43.67 | $-83.57 \pm 0.43$ | 68.87 | $29.07 \pm 0.48$ | 2.87 | $21.55 \pm 1.22$ | 19.49 |
| N-MC OI | $5.89 \pm 1.03$ | 4.28 | $-8.23 \pm 0.94$ | 6.54 | $25.62 \pm 0.51$ | 2.72 | $46.20 \pm 1.94$ | 5.55 |
| N-MC IPW | $5.62 \pm 0.85$ | 4.81 | $-7.10 \pm 0.94$ | 7.66 | $26.65 \pm 1.44$ | 3.19 | $46.52 \pm 1.92$ | 5.85 |
| N-OI | $4.84 \pm 1.19$ | 5.39 | $\mathbf{-16.57 \pm 1.06}$ | **2.15** | $29.05 \pm 1.77$ | **1.92** | $44.67 \pm 1.56$ | 3.99 |
| N-IPW | $\mathbf{9.06 \pm 0.69}$ | **1.26** | $-12.54 \pm 0.86$ | 2.33 | $25.64 \pm 0.40$ | 2.68 | $\mathbf{42.53 \pm 2.07}$ | **2.57** |
| Ground Truth | **10.11** [NCT03987919, 18] | | **-14.7** [NCT03191396, 8] | | **28.3** [NCT03828539, 37] | | **41.00** [NCT02191579, 39] | |

the other three as held-out test settings, see appendix C. We include further details for all our datasets in appendix E, including the definitions of covariates and outcomes.

Next, we investigate several questions about the performance of NATURAL empirically. We used GPT-4 Turbo for sampling and LLAMA2-70B for computing conditional probabilities.

**How well does NATURAL estimate observational distributions from self-reported data?** Our semi-synthetic datasets give us access to the true joint distributions $P(X, T, Y)$ and true propensity scores $P(T = 1|X)$. The top row of fig. 3 shows the KL divergence between these distributions and those estimated by NATURAL Full, for Hillstrom (left) and Retail Hero (right). We find that these KL divergences decrease steadily as the number of reports used in the estimation increases. The bottom row shows corresponding root-mean-squared error (RMSE) between NATURAL and the true ATE. This corroborates the insight that as the joint distribution and propensity scores are estimated more accurately, the predicted ATE gets closer to its true value. In particular, we observe a clear correlation between the quality of estimated propensity scores and estimated ATEs.

**How do NATURAL methods compare to one another and to trained baselines?** We present our estimated ATE and its RMSE on semi-synthetic datasets in table 1. Further, we evaluate two trained baselines, which use a Bag-of-Words model and a sentence encoder respectively, to train representations of text data with their labels. Here, for each attribute in the set of covariates, treatments, and outcomes, we train a MLP model with 5-fold cross validation to predict that attribute. We then use these predicted attributes as a tabular dataset of samples that can be plugged into any causal inference estimator. We find that our methods are competitive with or outperform these baselines, despite not being trained with any labels. In particular, the sentence encoder baseline collapsed to an ATE of zero for Hillstrom, having learned the constant predictor for outcomes. IPW (Structured) is an oracle estimator, which assumes full access to structured data. Selection-biased N-IPW demonstrates the challenge of ATE estimation in the presence of bias which was systematically simulated as a function of covariates "channel" and "zip code" for Hillstrom and "age" for Retail Hero.

Table 2 compares NATURAL methods to estimate the ATEs in real-world clinical settings using self-reported data from the PushShift collection of Reddit posts. Remarkably, our predicted ATEs (a) depict the same *direction of effect*, and (b) fall *within 3 percentage points* of their corresponding ground truth clinical trial ATEs. For both semi-synthetic and real data experiments, NATURAL IPW outperforms other versions across datasets, except for the Semaglutide vs. Liraglutide setting, where NATURAL OI performed the best. Both N-MC versions perform similarly on all datasets.

This result is significant. Clinical trials can take on the order of years and costs in the tens to hundreds of millions of dollars. Going from the raw language observational data to ATE in our framework takes on the order of days and costs at most a few hundred dollars of compute. For problems in medicine, economics, sociology, and political science where randomization is infeasible or expensive, NATURAL provides a tractable way to leverage observational data to rank potential experiments prior to conducting them.

**How do different choices in the NATURAL pipeline effect ATE prediction?** We assess the impact of key choices in our pipeline described in section 4, by ablating them one-by-one. We investigated and selected these choices on the Semaglutide vs. Tirzepatide experiment. Figure 4 (left) compares the RMSE of predicted ATEs when data is not filtered according to inclusion criteria and LLM imputations are replaced with samples from an uniform distribution.

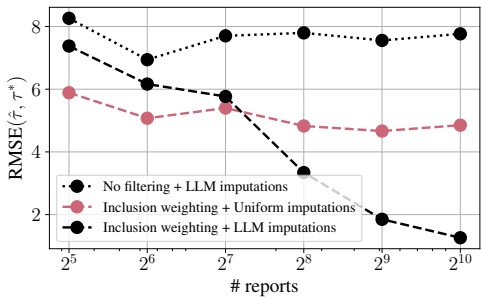 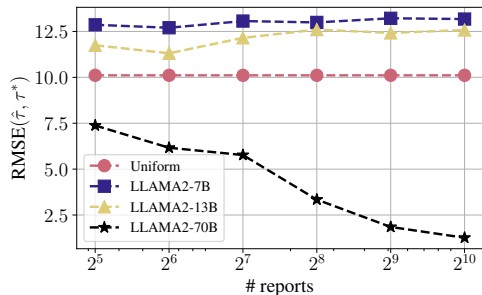

Figure 4: Ablation study on Semaglutide vs. Tirzepatide, to tease apart the effect of data filtering and imputation (left) as well as LLM scale for conditionals (right) on NATURAL performance.

It shows that both inclusion-based filtering and imputations from a pretrained LLM are crucial for the performance of NATURAL. We also compared performance of our method when the conditional probabilities in eq. (6) are evaluated using models of different scales in fig. 4 (right), and found that performance improves at larger scales and with greater quantity of data.

**How well do different estimates of propensity score balance covariates?** A property of accurate propensity score estimates is that they balance covariates across treatment cohorts (see Ding [11] for details and proofs), *i.e.* the average treatment effect on each covariate, corrected using propensity scores, is close to zero. Figure 5 visualizes this quantity for different covariates of the Semaglutide vs. Tirzepatide experiment and shows that propensity scores

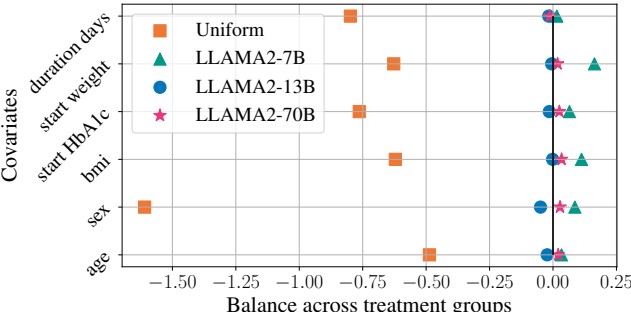

Figure 5: NATURAL propensity scores balance the Semaglutide vs. Tirzepatide covariates better than uniform scores.

estimated using LLAMA2 conditional distributions balance the covariates far better than a uniform distribution does, with the 70B model consistently estimating the treatment effect on each covariate as close to zero. Similar visualizations for the test settings are shown in fig. 10 of appendix F.2.

## 7 Conclusion

In this work, we introduced *NATURAL*, a family of text-conditioned estimators, to automate treatment effect estimation using free-form text data. We demonstrated NATURAL's efficacy with six semi-synthetic and real datasets for systematic evaluation of its pipeline. We exposed the ability of LLMs to extract meaningful conditional distributions over structured variables and how their combination with classical causal estimators can predict real-world causal effects with remarkable accuracy. Given this promising performance, exciting directions for future work include (i) incorporating automatic prompt tuning methods into the pipeline, (ii) extending our methods to real-valued $(X, T, Y)$, (iii) exploring whether our assumptions can be weakened, (iv) exploring other domains in applied research, *e.g.*, social sciences, (v) performing a more extensive evaluation of NATURAL on different study designs to better understand what type of treatments, outcomes, and reports show better or worse practical performance with NATURAL or (vi) deploying the pipeline to test hypotheses at even larger scales.

NATURAL estimators have numerous use cases with potentially far-reaching impact. As long as patients have access to treatments and report their experiences, NATURAL can be used to compare two treatments in new indications or new populations. Therefore, our pipeline can in principle support efforts to prioritize trials for repurposed drugs or supplements in under-served diseases or populations. Further, a crucial step after drug approval is post-marketing surveillance for side effects (positive or negative) that may not have been measured or may have been too rare to identify in a smaller trial. NATURAL can leverage the diversity of available language data to detect these effects. While our motivations largely stem from the challenges of drug development, our NATURAL estimators are applicable to any effect estimation setting for which there exists relevant natural language data.

## Acknowledgements

Our work was directly inspired by work done by Noah MacCallum, George Hosu, Sina Hartung, and Zain Memon at Eureka Health. Their project, Social Treatment Insights, explored the use of LLMs with social media data to draw medical insights. We thank them for the spark that led to this project and for the suggestion to study weight loss treatments. We would also like to thank Dexter Ju for useful practical suggestions on filtering social medial posts, Amol Verma and Fahad Razak for pointers on migraine-related clinical keywords, and David Lopez-Paz, Patrick Forré, Roger Grosse and Sheldon Huang for feedback on an initial draft of the paper. Resources used in preparing this research were provided in part by the Province of Ontario, the Government of Canada through CIFAR, and companies sponsoring the Vector Institute. We acknowledge the support of the Natural Sciences and Engineering Research Council of Canada (NSERC), RGPIN-2021-03445.

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

# Appendix

## A   Notation

| | |
|---|---|
| $R$ | Random variable corresponding to unstructured natural language text from a social media post (or report). |
| $X$ | Random variable corresponding to features of an individual in a causal inference dataset. |
| $T$ | Random variable corresponding to treatment or intervention assigned to an individual in a causal inference dataset. |
| $Y$ | Random variable corresponding to outcome observed for an individual in a causal inference dataset. |
| $x$ | Possible instance of $X$ from its support $\mathcal{X}$. |
| $t$ | Possible instance of $T$ from its support $\mathcal{T} = \{0, 1\}$ (binary treatments). |
| $y$ | Possible instance of $Y$ from its support $\mathcal{Y} = \{0, 1\}$ (binary outcomes). |
| $r$ | Possible instance of $R$ from its support $\mathcal{R}$. |
| $Y(t)$ | Random variable corresponding to potential outcome observed for an individual after receiving treatment $t$. |
| $e(X)$ | Propensity score function for binary treatments, equal to $P(T = 1 \mid X)$. |
| $X_i$ | Sampled value of $X$ for individual $i$. |
| $T_i$ | Sampled value of $T$ for individual $i$. |
| $Y_i$ | Sampled value of $Y$ for individual $i$. |
| $R_i$ | Sampled report $R$ for individual $i$. |
| $\tau$ | Average treatment effect (ATE) given by $\mathbb{E}[Y(1) - Y(0)]$, where the expectation is over some defined population of individuals. |
| $n$ | Total number of individuals. |
| $n_1$ | Total number of individuals that are assigned treatment $T = 1$. |
| $n_0$ | Total number of individuals that are assigned treatment $T = 0$. |

# B  Deriving NATURAL Full from Outcome Imputation estimator

In section 3, we derived the idealized version of our method, NATURAL Full using the form of the IPW estimator. In fact, any unbiased estimator would lead to same form as eq. (3), which can then be used to derive NATURAL Full using reports $R$ and the law of iterated expectations. Here, we show this using the Outcome Imputation estimator. For discrete $X$ and binary $T$ and $Y$, we have

$$\tau = \mathbb{E}_X[P(Y = 1 \mid T = 1, X) - P(Y = 1 \mid T = 0, X)] \qquad \text{(outcome imputation)}$$

$$\text{(13)}$$

$$= \sum_{x \in \mathcal{X}} P(X = x)[P(Y = 1 \mid T = 1, X = x) - P(Y = 1 \mid T = 0, X = x)] \qquad \text{(expectation of discrete } X\text{)}$$

$$\text{(14)}$$

$$= \sum_{x \in \mathcal{X}} P(X = x) \left[ \frac{P(Y = 1, T = 1, X = x)}{P(X = x)P(T = 1|X = x)} - \frac{P(Y = 1, T = 0, X = x)}{P(X = x)P(T = 0|X = x)} \right] \qquad \text{(expanding conditionals)}$$

$$\text{(15)}$$

$$= \sum_{x \in \mathcal{X}} \left[ \frac{P(Y = 1, T = 1, X = x)}{e(x)} - \frac{P(Y = 1, T = 0, X = x)}{1 - e(x)} \right] \qquad \text{(definition of } e(x)\text{)}$$

$$\text{(16)}$$

$$= \sum_{x \in \mathcal{X}} \sum_{y \in \mathcal{Y}} y \left[ \frac{P(Y = y, T = 1, X = x)}{e(x)} - \frac{P(Y = y, T = 0, X = x)}{1 - e(x)} \right] \qquad \text{(since } \mathcal{Y} = \{0, 1\}\text{)}$$

$$\text{(17)}$$

$$= \sum_{x \in \mathcal{X}} \sum_{y \in \mathcal{Y}} y \sum_{t \in \mathcal{T}} \left[ t \frac{P(Y = y, T = t, X = x)}{e(x)} - (1 - t) \frac{P(Y = y, T = t, X = x)}{1 - e(x)} \right] \qquad \text{(since } \mathcal{T} = \{0, 1\}\text{)}$$

$$\text{(18)}$$

$$= \sum_{(x,t,y)} P(Y = y, T = t, X = x) \left[ \frac{ty}{e(x)} - \frac{(1 - t)y}{1 - e(x)} \right] \qquad \text{(rearranging terms)}$$

$$\text{(19)}$$

$$= \mathbb{E}_{(X,T,Y)} \left[ \frac{TY}{e(X)} - \frac{(1 - T)Y}{1 - e(X)} \right], \qquad \text{(20)}$$

which is equivalent to eq. (3).

# C   Worked Example: Semaglutide vs. Tirzepatide

To make the NATURAL pipeline and its implementation more concrete, we now work through an end-to-end example using the Semaglutide vs. Tirzepatide dataset. We used this setting to develop our evaluation setup and constructed the pipeline as a function of the experiment design. While it is infeasible to exhaustively describe the entire decision space, major decisions that were found to impact ATE prediction are (i) filtering strategy, (ii) prompt tuning for extraction, and (iii) covariates' discretization. The strategies we tried are included in the pipeline description below, with the specific decisions made for each highlighted in this color.

**Pipeline for Semaglutide vs. Tirzepatide.**    Given the experimental design of the clinical trial in Frías et al. [18] (NCT03987919), we defined our experiment as follows:

1. Treatments: Semaglutide and Tirzepatide
2. Outcome: Percentage of participants who lost 5% or more of their initial weight
3. Covariates: Age, Sex, BMI, Start weight, Start HbA1c, Duration
4. Inclusion criteria:
   (a) The user must be diagnosed with Type 2 Diabetes with starting HbA1c between 7% and 10.5%.
   (b) They must already be on a regime of the treatment called Metformin.
   (c) They must have a BMI of 25 kg/m$^2$ or more.
   (d) Since different treatment dosages can have varying effects, we also included dosage as an inclusion criterion, *i.e.* we aimed to include only posts that reported taking 1mg for Semaglutide and 5mg for Tirzepatide, as in the clinical trial.

Decisions in the pipeline that are a direct function of an experimental design like the one above are highlighted in this color in the following description.

Next, we implemented the entire pipeline as follows:

1. **Initial filter.** We identified nine subreddits relevant to this problem setting: r/Mounjaro, r/Ozempic, r/fasting, r/intermittentfasting, r/keto, r/loseit, r/Semaglutide, r/SuperMorbidlyObese, r/PlusSize. From each subreddit, we downloaded all submissions and comments posted upto December 2022 from the PushShift collection, so as to only use publicly available data. This resulted in a dataset of 577,733 submissions and comments. An initial deterministic, task-agnostic and rule-based filter removed any submission or comment if its content was not a string, if it had no score, if the content was `"[deleted]"` or `"[removed]"`, if it was a comment with fewer than ten space-separated strings (presumably, words), if the author's name contained the string `"bot"`, if there were no spaces in the first 2048 characters, and if less than 50% of all characters were alphabetic. This reduced the dataset size to 380,276. We then formatted this data into dictionary-like datapoints with fields: `subreddit`, `title`, `date created`, `post/comment`, `author replies`. We indcluded the last field because comments written by the author as replies to their own post may contain additional relevant information when combined with with original post and other replies. We then passed these through a task-dependent string-matching filters. For this dataset in particular, we listed strings used commonly to refer to the treatments, `["ozempic", "mounjaro", "semaglutide", "tirzepatide", "wegovy", "rybelsus", "zepbound"]`, included common misspellings generated with GPT-4 and Perplexity, and filtered out datapoints that did not contain any of these strings. Similarly, we listed keywords relevant to the outcome of interest, `["kg", "kilo", "lb", "pound", "weigh", "drop", "loss", "lost", "gain", "hb", "a1c", "hemoglobin", "haemoglobin", "glucose", "sugar"]` and filtered out datapoints that did not contain any of these strings. This filtered dataset now contained 50,654 datapoints.
2. **Filter by relevance.** Next, we wrote a problem setting description and prompted GPT-3.5 Turbo to determine whether the posts, along with auxiliary information from the formatted dictionaries described above, were relevant to the described setting. The description and instructions for this particular dataset are shown in prompt 2. We manually labeled a handful of datapoints as `Yes` or `No` and included these as incontext examples to improve the LLM's generations. We removed datapoints that were deemed irrelevant, resulting in a "relevant" dataset of 21,229 datapoints.
3. **Filter by treatment-outcome.** To further filter the data to points that refer specifically to the treatments and outcome of interest, we prompted GPT-3.5-Turbo to extract only information

required to ascertain the treatment and outcome, as shown in prompt 3. Since the outcome for this dataset, achievement of a target weight loss of 5% or more, may be reported in several ways, we attempted to cover all those possibilities. Specifically, we prompted the LLM to extract the user's starting weight, end weight, change in weight and percentage of change in weight. Several combinations of these attributes allow us to programmatically infer the final outcome. We also extracted the units in which weight was reported, converting all extractions to be in lbs. We filtered out any datapoint for which the extracted treatment was not one of the treatments considered for this task or for which it was not possible to infer the outcome using the above-mentioned extracted information. This finally gave us a natural language dataset of 4619 relevant reports, each of which contained treatment and outcome information pertaining to the defined problem setting.

4. **Filter known covariates by inclusion criteria.** To evaluate against the clinical trial, we further filtered the dataset according to its inclusion criteria. At a high level, different strategies for this filtering trade-off how strictly we match the criteria with how many datapoints remain in the filtered set. Strict filtering to match every criteria exactly resulted in very few reports. Less strict filtering to match a subset of the criteria resulted in predicted ATEs that varied depending on which criteria we chose to satisfy, with no apparent task-agnostic principle to determine crucial criteria. Finally, we used instructions shown in prompt 4 to extract covariates and aimed for a set of reports with non-zero probability of satisfying these criteria. As described in item (iv) of section 4 and motivated in appendix G, we extracted all covariates including ones related to the inclusion criteria and removed datapoints whose extractions were not Unknown and failed to satisfy the criteria above. This resulted in a dataset of 1265 reports.

5. **Extract known and unknown covariates.** Treating these 1265 reports as the final dataset from which to estimate an ATE, we extracted the set of covariates given in our experiment definition. We also included the duration of treatment as a covariate since this information is often reported and is likely to influence the outcome. This extraction step was conditioned on inclusion criteria being satisfied, a description of which was included in the extraction prompt, as in prompt 5. We tuned the prompts for extracting attributes, which include general instructions for the task and specific questions for each attribute. This was done by inspecting a handful of reports and corresponding extractions and then modifying prompts to correct any observed errors.

6. **Infer conditionals.** We inferred conditional distributions from LLAMA2-70B for different versions of NATURAL, with the strategy described in item (vi) of section 4 and LLM inputs of the form shown in prompt 6. Here, "conditioning on covariates" was implemented by adding questions about the covariates and their sampled answers to the input. For instance, for sex, the question "What is the reported sex of the user?" was followed by its previously extracted answer (Male or Female). The scoring strategy required enumerating possible options for treatments and outcomes for each input, which were ["Semaglutide like Ozempic or Wegovy or Rybelsus", "Tirzepatide like Mounjaro or Zepbound"] and ["No", "Yes"], respectively.

7. **Weight reports according to inclusion criteria match.** We also used LLAMA2-70B to compute the weighting terms described in item (vii) of section 4. Concretely, we constructed a prompt, like prompt 7, describing the inclusion criteria listed in item 4 of the experiment design above, followed by a report, $R_i$ and an instruction asking the LLM to determine whether all the described criteria are met. We then scored the possible answers, ["No", "Yes"], exponentiated and renormalized them to obtain $P(X \in I \mid R_i)$. We marginalized over reports to compute the denominator, $P(X \in I)$, in the weight. The contribution of each report to the ATE estimates was weighted by this relative likelihood of matching the in inclusion criteria of the experiment given the report.

8. Finally, given all the required extractions and conditional probabilities, we required discrete covariates to plug them into our NATURAL estimators. Hence, we converted any continuous covariates into discrete categories. These categories for each dataset are shown in table 4 for all our datasets. Different choices of discretization led to slightly different ATE predictions. We found it most helpful to discretize continuous numerical covariates into intervals such that the number of datapoints were roughly balanced across intervals. This avoided covariate strata with too many or too few datapoints and resulted in ATE predictions from all NATURAL estimators that were sufficiently close to the ground truth.

**Adapting the pipeline to test trials.** The decisions above in this color directly or indirectly influenced the ATE and we made our choices with access to the ground truth for Semaglutide vs. Tirzepatide. Hence, we call this a "tuned" setting. We fixed these decisions for the three test settings. Note that no other aspect of the pipeline depends on the ATE. All the choices in this color are a

function of the experimental design. Hence, the pipeline can be easily adapted to any new setting, given its experimental design and without knowledge of the true ATE.

# D LLM Prompts

**Prompt 1: Semi-synthetic report generation (Hillstrom)**

```
You are a user who used a website for online purchases in the past one year
    and want to share your background and experience with the purchases on
    social media.

## Attributes
The following are attributes that you have, along with their descriptions.
> {features}

## Personality Traits
The following dictionary describes your personality with levels (High or Low)
    of the Big Five personality traits.
> {traits}

## Your Instructions
Write a social media post in first-person, accurately describing the
    information provided. Write this post in the tone and style of someone
    with the given personality traits, without simply listing them.
Only return the post that you can broadcast on social media and nothing more.

## Post
>
```

**Prompt 2: Relevance filtering (Weight Loss)**

```
You are an expert researcher looking around reddit for posts/comments
    describing the effect of a treatment on weight loss or blood sugar level
    experienced by the author.

## Problem Setting
> You are interested in self-reported effects of a treatment on a user who
    took the treatment themselves. You want to be able to answer some or all
    of the following questions from the text of the post or comment:
1. Which treatment did the user take?
2. What change did they observe in their weight due to this treatment, and
    during what duration did they observe this change?
3. What change did they observe in their blood sugar, aka HbA1c levels, due
    to this treatment, and during what duration did they observe this change?
4. What are other attributes they report, e.g. age, sex, country of
    residence, diabetes diagnosis, other treatments they have tried, or side
    effects?

## Your Instructions
I will show you a post or comment, and contextual information about it. Based
    on the given problem setting and contextual information, you need to
    judge whether it is relevant to the problem setting described above or
    not. Answer Yes if the post is relevant and No otherwise; nothing else.
Here are a few examples:

{incontext examples}

## Subreddit
> This post was found on the subreddit r/{subreddit}.

## Title
> This post was titled: {title}

## Date Created
> This post was created on {date_created}.

## Post
> {post}

The author also replied with the following in the thread:
> {replies}

Answer Yes if the comment is relevant and No otherwise, and nothing more.
## Your Answer
>
```

**Prompt 3: Treatment-outcome filtering (Weight Loss)**

You are a medical assistant, helping a doctor structure posts about weight
    loss treatments found on Reddit. Your task is to use the self-report to
    interpret accurate information about the following fields and store them
    in a JSON dictionary.

## Your Instructions
I will provide a post along with its subreddit name, title and date of
    creation. You must return a valid JSON dictionary containing the
    following keys along with the corresponding accurate information:
"start_weight": Numerical value for the user's starting weight, before
    starting the treatment described, sometimes referred to as SW.
"end_weight": Numerical value for the user's current or final weight, at the
    end of the treatment regime, sometimes referred to as CW.
"weight_unit": Units in which weight is reported: "kg" or "lb".
"weight_change": Numerical value for net change in the user's weight. Use a
    postive sign to indicate weight gain and negative sign for weight loss.
    Leave blank if it is not possible to infer the change in weight.
"percentage_weight_change": Numerical value for percentage reduction in
    user's weight relative to their start weight. Use a postive sign to
    indicate weight gain and negative sign for weight loss. Leave blank if
    it is not possible to infer the percentage.
"drug_type": Treatment taken by the user: "Semaglutide", "Tirzepatide" or
    "Other". Semaglutide includes Ozempic, Wegovy or Rybelsus. Tirzepatide
    includes Mounjaro or Zepbound.

Assign a valid value to each key above. If you can't find the required
    information in the post, assign the value "Unknown". Remember to ONLY
    return a valid JSON with ALL of the above keys and their accurate values.

**Prompt 4: Covariate extraction (Weight Loss)**

As a medical assistant aiding a physician, your role involves examining
    Reddit posts discussing weight loss treatments and interpreting
    self-reported information accurately. This data needs to be translated
    into a well-structured JSON dictionary, with the most suitable option
    chosen from the choices provided.

## Your Instructions
Assume a user shares a post along with related data. Your job will be to
    create a dictionary comprising of the following keys as well as their
    matching accurate data:

{covariate descriptions}

Please ensure you fill all the fields and that you choose a valid value for
    each key from the provided options. Unfilled fields are not allowed. In
    instances where certainty is impossible, make your best educated guess,
    or provide the "Unknown" value. Note that your completed task should
    ONLY yield a JSON containing ALL the listed keys alongside their
    accurate values.

Here are a few examples:

{incontext examples}

## Input
{report}

## Output
>

**Prompt 5: Covariate imputation (Weight Loss)**

```
You are a medical assistant tasked with creating a profile of a patient who
    is taking a weight loss treatment, and presenting it as a JSON
    dictionary with prespecified keys. Fill in suitable values for ALL the
    keys. You can use information provided about the patient.

## Your Instructions
A patient has Type 2 Diabetes, is known to have taken Metformin for the last
    3 months and has a BMI greater than 25 kg per meter squared.
Dosage for Semaglutide, Ozempic, Wegovy and Rybelsus is 1mg. Dosage for
    Tirzepatide, Mounjaro and Zepbound is 5mg.
Create a possible profile for this patient with the following fields and
    represent it as dictionary:

{covariate descriptions}

Please ensure you fill all the fields with a valid value. Unfilled fields or
    values like "Unknown" are not allowed. Note that your completed task
    should ONLY yield a JSON containing ALL the listed keys alongside their
    accurate values.

Here is an entry that the patient wrote about themselves, which may be useful
    for your task.
## Input
{report}

## Output
>
```

**Prompt 6: Conditional distribution inference (Weight Loss)**

```
You are a medical assistant aiding a physician. I am going to ask you a few
    multiple choice questions about some posts I just found online. Please,
    answer accordingly.

## Your Instructions
I will give you a post about an individual's experience with a treatment and
    its effect on their weight, and a few questions with their correct
    answers, followed by additional multiple choice questions and options to
    choose from. Pick the right answer.

## Social Media Post
> {report}

## Questions and their correct answers
Q: {question about covariate X1} A: {X1 sample}.
Q: {question about covariate X2} A: {X2 sample}.
..

## Questions
Q: Which treatment did the user take?
Options: a) {t0} b) {t1}
A: {t0}

Q: Did the user lose 5 or more percent of their initial weight?
Options: a) {y0} b) {y1}
A: {y0}
```

**Prompt 7: Inclusion weights (Weight Loss)**

You are a medical assistant aiding a physician. Based on the following social
    media post about an individual's experience with a diabetes treatment
    and its effect on their weight, evaluate whether the person meets ALL of
    the following criteria:

1. Type 2 Diabetes: Diagnosed with Type 2 Diabetes and has an HbA1c between
    7\% and 10\%.
2. Metformin: Has been taking Metformin for at least the past 3 months.
3. BMI: Has a BMI greater than 25 kg/m^2.
4. Medication Dosage: If taking Semaglutide (e.g., Ozempic, Wegovy,
    Rybelsus), the dosage is 1mg; OR if taking Tirzepatide (e.g., Mounjaro,
    Zepbound), the dosage is 5mg.

After analyzing the post, determine whether the individual meets ALL of the
    above criteria.

Social Media Post
> {report}

## Question
Q: Does the user satisfy the given inclusion criteria?
Options: a) No b) Yes
A: {No/Yes}

# E  Dataset Details

We provide further details about the treatments, outcomes and covariates, along with inclusion criteria and discrete categories used in our experiments, for each dataset in tables 3 and 4.

Table 3: Treatments, outcomes and synthetic confounders (where applicable) for each dataset.

| Dataset | Treatment | Outcome | Synthetic confounder |
|---|---|---|---|
| **Hillstrom** | email communication | website visit | newbie |
| **Retail Hero** | SMS communication | purchase | age |
| **Semaglutide vs. Tirzepatide** | corresponding drug | weight loss of 5% or more | NA |
| **Semaglutide vs. Liraglutide** | corresponding drug | weight loss of 10% or more | NA |
| **Erenumab vs. Topiramate** | corresponding drug | discontinuation due to adverse effects | NA |
| **OnabotulinumtoxinA vs. Topiramate** | corresponding drug | discontinuation due to adverse effects | NA |

Table 4: Covariate descriptions, corresponding discrete categories and inclusion criteria enforced for each dataset. Intervals for continuous numerical variables were determined from the extracted values such that each discrete category is roughly balanced in terms of its number of datapoints.

| Covariate | Description | Discrete categories | Inclusion criteria |
|---|---|---|---|
| **Hillstrom** | | | |
| recency | number of months since last purchase | $[1-4, 5-8, 9-12]$ | |
| history | dollar value of previous purchase | $[0-100, 100-200, ..., > 1000]$ | |
| mens | purchase of men's merchandise | [True,False] | |
| womens | purchase of women's merchandise | [True,False] | NA |
| zip_code | type of area of residence | [Suburban area,Rural area,Urban area] | |
| newbie | new customer | [True,False] | |
| channel | channel used for purchases | [Phone,Web,Multichannel] | |
| **Retail Hero** | | | |
| avg. purchase | avg. purchase value per transaction | $[1-263, 264-396, 397-611, > 612]$ | |
| avg. product quantity | avg. number of products bought | $[\leq 7, > 7]$ | |
| avg. points received | avg. number of points received | $[\leq 5, > 5]$ | NA |
| num transactions | total number of transactions so far | $[\leq 8, 9-15, 16-27, > 28]$ | |
| age | age of user | $[\leq 45, > 45]$ | |
| **Semaglutide vs. Tirzepatide** | | | |
| age | age of user | $[\leq 45, > 45]$ | |
| sex | sex of user | [Male,Female] | (t2 diabetes==True) |
| bmi | body mass index of user | $[\leq 28.5, > 28.5]$ | & ($7 \leq$ start HbA1c $\leq 10.5$)) |
| start HbA1c | initial glycated haemoglobin value | $[\leq 7.5, > 7.5]$ | & (metformin==True) |
| start weight | initial weight in lbs | $[\leq 220, > 220]$ | & (bmi $\geq 25$) |
| duration (days) | number of days treatment was taken for | $[\leq 90, > 90]$ | |
| **Semaglutide vs. Liraglutide** | | | |
| age | age of user | $[\leq 45, > 45]$ | |
| sex | sex of user | [Male,Female] | (t2 diabetes==True) |
| bmi | body mass index of user | $[\leq 28.5, > 28.5]$ | & ($7 \leq$ start HbA1c $\leq 11$)) |
| start HbA1c | initial glycated haemoglobin value | $[\leq 7.5, > 7]$ | & (metformin/other==True) |
| start weight | initial weight in lbs | $[\leq 220, > 220]$ | |
| duration (days) | number of days treatment was taken for | $[\leq 120, > 120]$ | |
| **Erenumab vs. Topiramate** | | | |
| age | age of user | $[\leq 32, > 32]$ | |
| sex | sex of user | [Male,Female] | ($18 \leq$ age $\leq 65$) |
| country | country of residence | [United States,Canada,...] | & (pregnant==False) |
| baseline MMD | initial number of monthly migraine days | $[\leq 6, > 6]$ | & (baseline MMD $\geq 4$) |
| duration (days) | number of days treatment was taken for | $[\leq 30, > 30]$ | |
| **OnabotulinumtoxinA vs. Topiramate** | | | |
| age | age of user | $[\leq 25, > 25]$ | |
| sex | sex of user | [Male,Female] | ($18 \leq$ age $\leq 65$) |
| country | country of residence | [United States,Canada,...] | & (baseline MMD $\geq 15$) |
| baseline MMD | initial number of monthly migraine days | $[\leq 15, > 15]$ | |
| duration (days) | number of days treatment was taken for | $[\leq 30, > 30]$ | |

# F    Further Experimental Results

## F.1    Known and Unknown/Imputed covariates for real data experiments

We refer the reader to figs. 6 to 9 for empirical distributions of covariates extracted by an LLM in its first extraction as well as those imputed in its second imputataion conditioned on inclusion criteria.

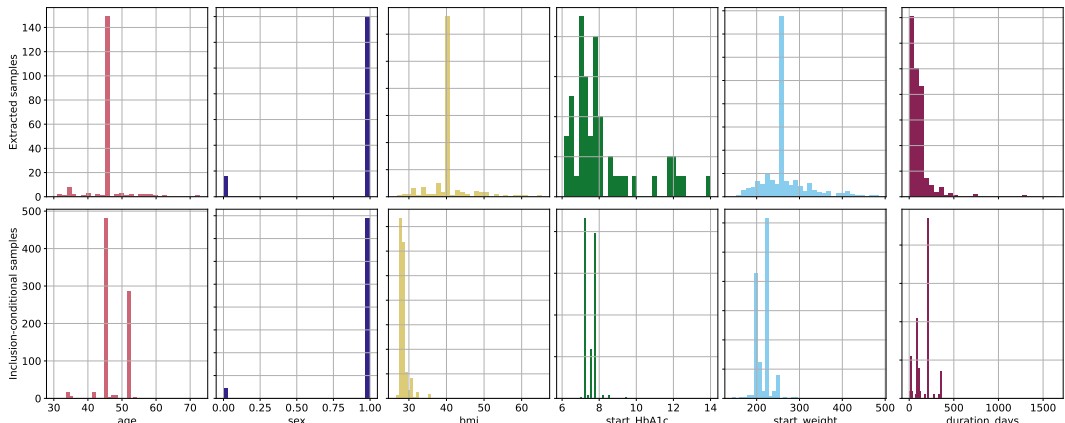

Figure 6: Distributions of "known" (top) vs "unknown" and imputed (bottom) covariates for Semaglutide vs. Tirzepatide.

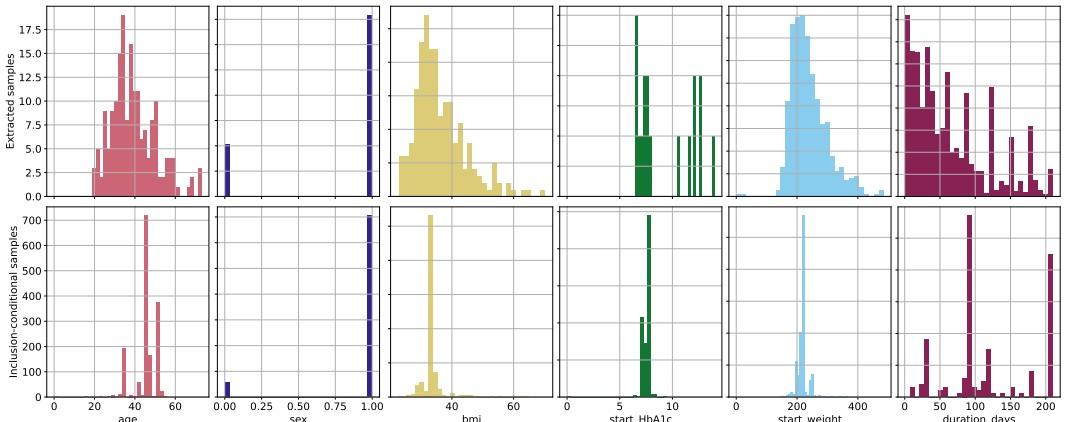

Figure 7: Distributions of "known" (top) vs "unknown" and imputed (bottom) covariates for Semaglutide vs. Liraglutide.

## F.2    Balancing property of propensity scores

We refer the reader to Figure 10 for visualizations of the propensity score corrected average treatment effect on covariates for all test clinical settings. For each setting, our estimated propensity score balances each covariate, far better than a uniform propensity distribution would.

Since covariates may take values at different scales, we computed the standard mean difference (SMD) across cohorts for each covariate $X^{(i)}$ [16], given by:

$$SMD = \frac{X^{(i)}(1) - X^{(i)}(0)}{\sqrt{0.5 * (\mathtt{var}(X^{(i)}(1)) + \mathtt{var}(X^{(i)}(0)))}}, \tag{21}$$

where $X^{(i)}(1) - X^{(i)}(0)$ estimates the average treatment effect on $X^{(i)}$, using propensity score weighting, and $\mathtt{var}(\cdot)$ denotes sample variance.

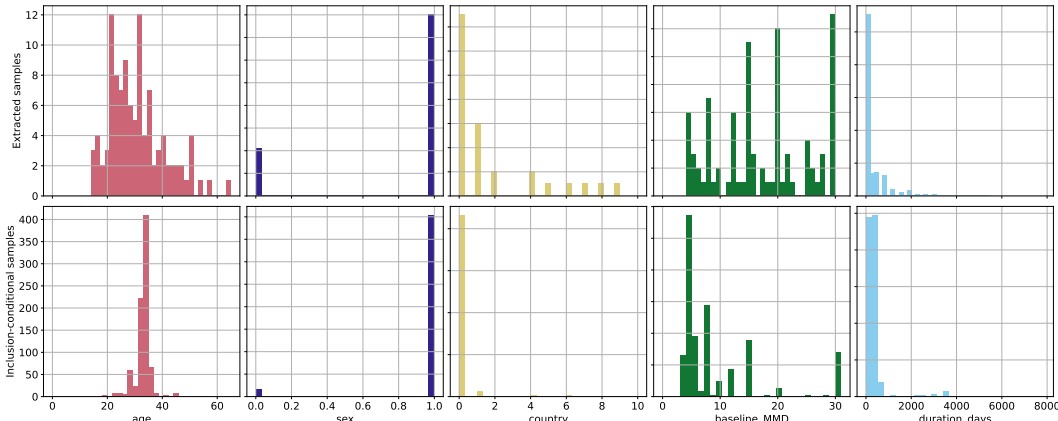

Figure 8: Distributions of "known" (top) vs "unknown" and imputed (bottom) covariates for Erenumab vs. Topiramate.

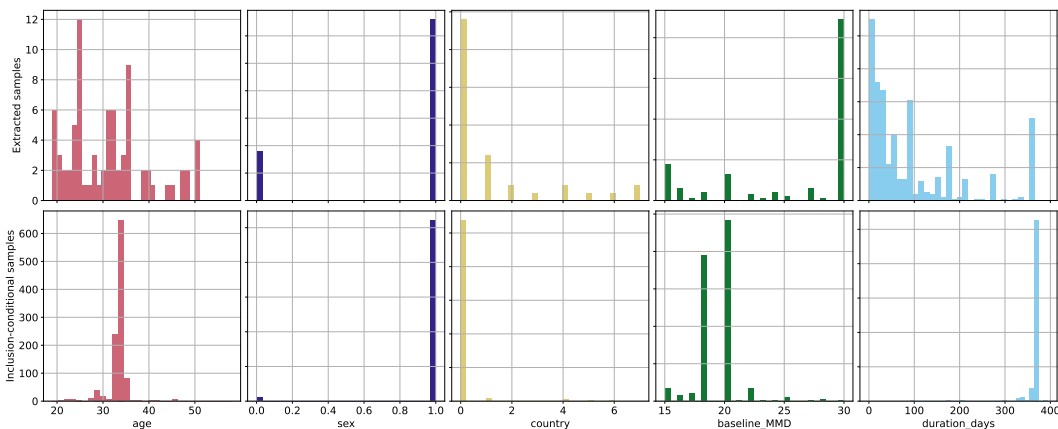

Figure 9: Distributions of "known" (top) vs "unknown" and imputed (bottom) covariates for Onabo-tulinumtoxinA vs. Topiramate.

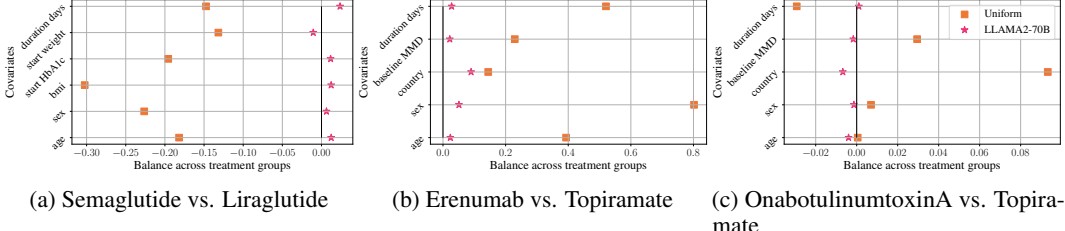

(a) Semaglutide vs. Liraglutide  (b) Erenumab vs. Topiramate  (c) OnabotulinumtoxinA vs. Topira-mate

Figure 10: Propensity scores estimated with LLAMA2-70B balance covariates of the real clinical datasets, far better than uniform scores do.

### F.3  Sensitivity Analysis

While we rely on domain expertise to define the confounder set for each setting such that necessary causal assumptions are satisfied, unobserved confoundedness remains a challenge. We followed strategies in Lu and Ding [29] to analyze the sensitivity of our ATE estimates to the degree of unobserved confoundedness. Specifically, the idea is to introduce sensitivity parameters,

$$\varepsilon_0(X) = \frac{\mathbb{E}[Y(0)|T=1,X]}{\mathbb{E}[Y(0)|T=0,X]} \text{ and } \varepsilon_1(X) = \frac{\mathbb{E}[Y(1)|T=1,X]}{\mathbb{E}[Y(1)|T=0,X]}$$

,

which quantify the degree of unobserved confoundedness. In our case, sensitivity parameters are the density ratio between likelihood of each potential outcome in the treated vs. untreated group. The ATE estimate is non-increasing in the sensitivity parameters. So, for positive ATEs, we are looking for the largest sensitivity parameters that maintain the positivity of the ATE, which tells us the degree of unobserved confoundedness that an estimator is robust to. Table 5 shows that the direction (sign) of

Table 5: NATURAL IPW for Semaglutide vs. Tirzepatide is robust to the degree of unobserved confoundedness shown below.

| $\varepsilon_1$ $\varepsilon_0$ | 1.00 | 1.05 | 1.10 | 1.15 | 1.20 | 1.25 |
|---|---|---|---|---|---|---|
| 1.00 | 9.45 | 6.51 | 3.85 | 1.41 | **-0.83** | **-2.88** |
| 1.05 | 8.51 | 5.58 | 2.91 | 0.47 | **-1.76** | **-3.82** |
| 1.10 | 7.58 | 4.64 | 1.97 | **-0.46** | **-2.70** | **-4.75** |
| 1.15 | 6.64 | 3.71 | 1.04 | **-1.40** | **-3.63** | **-5.69** |
| 1.20 | 5.71 | 2.77 | 0.10 | **-2.33** | **-4.57** | **-6.62** |
| 1.25 | 4.77 | 1.84 | **-0.83** | **-3.27** | **-5.50** | **-7.56** |

NATURAL IPW ATE estimates for the Semaglutide vs. Tirzepatide dataset change from positive to negative at large values of sensitivity parameters, implying that they are robust to large degrees of unobserved confoundedness.

We further investigated the importance of each confounder as suggested by the leave-one-covariate-out approach in Section 4 of Lu and Ding [29], by dropping that covariate as if it is an unobserved confounder and measuring the corresponding worst-case (over all possible values of the remaining covariates) sensitivity parameters with the remaining covariates. Figure 11 shows contour lines depicting NATURAL IPW ATE estimates at different values of sensitivity parameters for all our real-world datasets. Orange regions denote sensitivity parameter values for which the sign of the estimate ATE does not change, while blue regions denote the values that flip the direction of the estimate. Stars show sensitivity parameters for each covariate in the different settings. Hence, this estimator is sensitive to the covariate set and all covariates are important in the case of Semaglutide vs. Tirzepatide, while most are important for Erenumab vs. Topiramate. The direction of ATE estimates is not sensitive to covariates in the case of Semaglutide vs. Liraglutide and OnabotulinumtoxinA vs. Topiramate, implying that the estimated causal effects could only be explained away by an unobserved confounder that is stronger than all observed confounders.

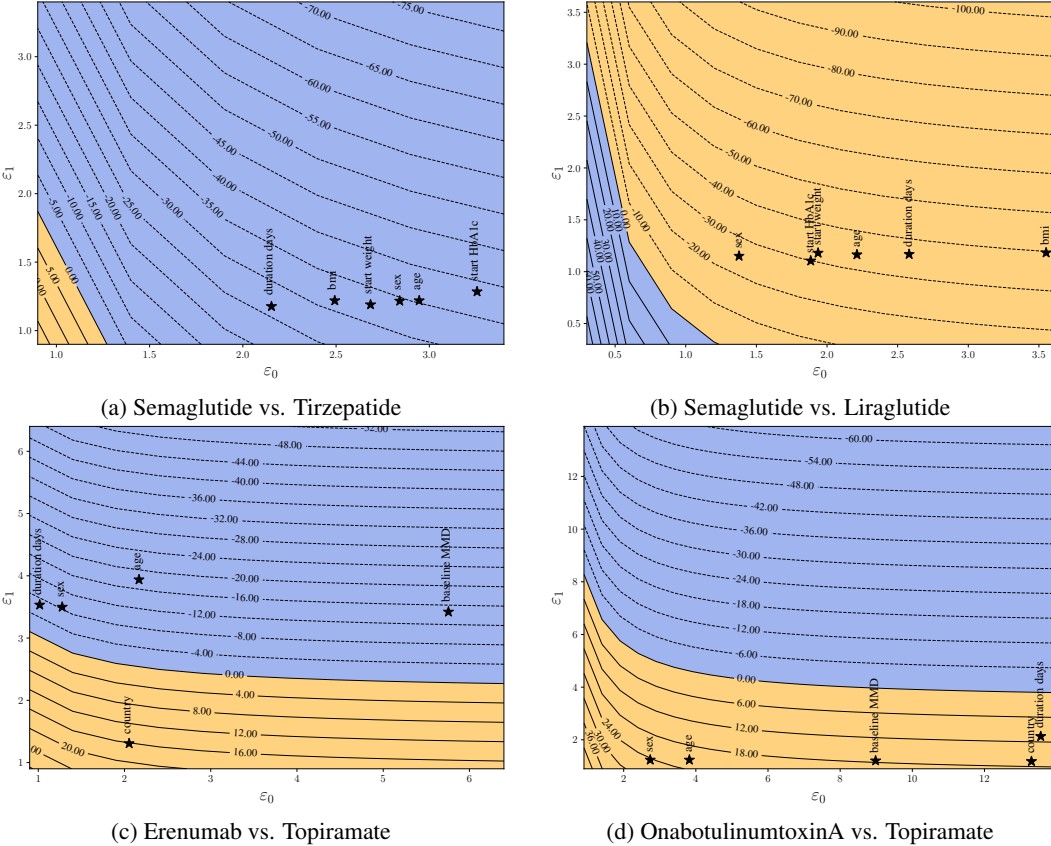

(a) Semaglutide vs. Tirzepatide

(b) Semaglutide vs. Liraglutide

(c) Erenumab vs. Topiramate

(d) OnabotulinumtoxinA vs. Topiramate

Figure 11: NATURAL IPW ATE estimates for different values of sensitivity parameters in each real setting. The orange (blue) region corresponds to ATEs with the same (flipped) sign as our estimates.

# G   Inclusion Criteria conditioned Estimator

We are interested in an ATE conditioned on inclusion criteria denoted $I$,

$$\tau(I) = \mathbb{E}[Y(1) - Y(0) \mid X \in I]. \tag{22}$$

Let $\tau(X, T, Y)$ be a function such that

$$\tau(I) = \mathbb{E}_{X,T,Y}[\tau(X, T, Y) \mid X \in I]. \tag{23}$$

For example, $\tau(I)$ can estimated by the IPW estimator,

$$\tau(X, T, Y) = \frac{TY}{e(X)} - \frac{(1-T)Y}{1 - e(X)},$$

because the $P(T = 1 | X = x, X \in I) = P(T = 1 | X = x)$ for all $x \in I$. Throughout this section, we operate under Assumptions 3 and 4 and assume that the LLM gives us access to the true data-generating conditionals.

The law of total expectation gives us an estimator that can operate on samples of reports $R$:

$$\begin{aligned}
\tau(I) &= \mathbb{E}_{X,T,Y}[\tau(X, T, Y) \mid X \in I] \\
&= \mathbb{E}_{R|X \in I}[\mathbb{E}_{X,T,Y}[\tau(X, T, Y) \mid X \in I, R]] \\
&= \sum_r P(R = r | X \in I) \mathbb{E}_{X,T,Y}[\tau(X, T, Y) \mid X \in I, R] \\
&= \sum_r P(R = r) \frac{P(X \in I | R = r)}{P(X \in I)} \mathbb{E}_{X,T,Y}[\tau(X, T, Y) \mid X \in I, R = r] \\
&= \mathbb{E}_R\left[\frac{P(X \in I | R)}{P(X \in I)} \mathbb{E}_{X,T,Y}[\tau(X, T, Y) \mid X \in I, R]\right].
\end{aligned}$$

To summarize, we have the identities:

$$\tau(I) = \mathbb{E}_{R|X \in I}[\mathbb{E}_{X,T,Y}[\tau(X, T, Y) \mid X \in I, R]] \tag{24}$$

$$= \mathbb{E}_R\left[\frac{P(X \in I | R)}{P(X \in I)} \mathbb{E}_{X,T,Y}[\tau(X, T, Y) \mid X \in I, R]\right]. \tag{25}$$

We prompted LLAMA2-70B with descriptions of the inclusion criteria and each report to estimate $\frac{P(X_i \in I | R_i)}{P(X_i \in I)}$, similar to other conditional distributions described in section 4, and marginalized over reports for the denominator.

It is also possible to avoid this weight computation at the expense of additional structural assumptions on the data. We discuss these conditions below and show corresponding results for NATURAL estimators in table 6.

Let $X \in \mathbb{R}^D$ and let us make the following assumption on the inclusion criteria:

**Assumption 5 (Inclusion criteria specification)** *The inclusion criterion $I$ defines a box, i.e., it is specified separately for each covariate dimension $I^d, d \in \{1, \ldots, D\}$ and the set of covariates satisfying every inclusion criteria is given by the product of individual criteria over the covariate dimensions, i.e., $\{X \in I\} = \prod_{d=1}^D \{X^d \in I^d\}$ where $X^d$ is the d-th dimension of $X = (X^d)_{d=1}^D$.*

Recall from section 4 that inclusion-based filtering leaves us with reports whose covariates are either "known" and satisfy their criteria or `Unknown`. We also have the value of the known covariates. Let $K \in \{0, 1\}^D$ be the binary vector of variables $K^d$ that indicate whether the covariate $X^d$ is found to be "known" for a random report $R$. Let $X^K = (X^d : K^d = 1)$ be the vector of length $\sum_d K^d$ holding the values of the known covariates. For ease of notation, define the event that the known covariates satisfy their criteria and the event that the unknown covariates satisfy their criteria:

$$\{X^K \in I^K\} = \{X^d \in I^d, \forall d: K^d = 1\} \tag{26}$$

$$\{X^{1-K} \in I^{1-K}\} = \{X^d \in I^d, \forall d: K^d = 0\} \tag{27}$$

Notice that $\{X^K \in I^K\} \cap \{X^{1-K} \in I^{1-K}\} = \{X \in I\}$. Thus, after the filtering steps we have

$$\{R_i, K_i, X_i^{K_i}\}_{i=1}^n \tag{28}$$

with the guarantee that the knowns satisfy their inclusion criteria, $\{X_i^{K_i} \in I^{K_i}\}$. Formally, assuming that the LLM computes the true conditional distribution of the data-generating process (Assumption 4), this gives us data sampled i.i.d. from $P(R = r, K = k, X^k = x^k | X^K \in I^K)$. Note, that we are assuming the existence of an additional ground-truth random variable $K$ in the data-generating process that describes whether a covariate is knowable from a report. Here, we show how to estimate $\tau(I)$ from this dataset of filtered reports using importance sampling, under the following assumption:

**Assumption 6 (Satisfaction of $I$ by `Unknown` covariates)** *Satisfaction of inclusion criteria by unknown covariates is conditionally independent of the report and the known covariates given satisfaction of inclusion criteria by known covariates, i.e., for all $r, k, x^k$:*

$$P(R = r, K = k, X^k = x^k | X^K \in I^K, X^{1-K} \in I^{1-K}) = P(R = r, K = k, X^k = x^k | X^K \in I^K) \tag{29}$$

One can derive the following identity in a similar fashion as eq. (25)

$$\tau(I) = \mathbb{E}_{R,K,X^K | X^K \in I^K} \left[ \frac{P(R, K, X^K | X \in I)}{P(R, K, X^K | X^K \in I^K)} \mathbb{E}_{X,T,Y}[\tau(X, T, Y) \mid X \in I, R, K, X^K] \right]. \tag{30}$$

From assumption 6, the fraction above simplifies to 1, leaving us with the following estimator

$$\tau(I) = \frac{1}{n} \sum_{i=1}^n \mathbb{E}_{X_i, T_i, Y_i}[\tau(X_i, T_i, Y_i) \mid X_i \in I, R_i, K_i, X_i^{K_i}], \tag{31}$$

which can be computed from the information available at the end of filtering. In practice, we do not condition the LLM on $K_i$ in the final inference step (vi), which amounts to an additional conditional independence assumption:

**Assumption 7 (Conditional independence of knowable covariates)** $K \perp\!\!\!\perp (T, Y) \mid (X, R)$.

Equation (31) above can now be more efficiently estimated by prompting the LLM to extract covariates under the constraints of the inclusion criteria for each report in our filtered dataset, and then following the remaining steps in the pipeline to an ATE estimate.

Table 6: ATE estimates on real datapoints that are filtered according to inclusion criteria but not weighted by the relative likelihood that they meet the inclusion criteria of the experiment given the report. Best performing NATURAL estimators fall within 3 percentage points of their corresponding ground truth clinical trial ATEs. Possible ATE values lie between $-100$ and $100$.

| | Tuned | | | | Held-out | | | |
|---|---|---|---|---|---|---|---|---|
| | Semaglutide vs. Tirzepatide (% weight loss $\geq$ 5%) | | Semaglutide vs. Liraglutide (% weight loss $\geq$ 10%) | | Erenumab vs. Topiramate (% discontinued due to AE) | | OnabotulinumtoxinA vs. Topiramate (% discontinued due to AE) | |
| | ATE (%) | RMSE | ATE (%) | RMSE | ATE (%) | RMSE | ATE (%) | RMSE |
| Uncorrected | $-33.56 \pm 0.77$ | 43.67 | $-83.57 \pm 0.43$ | 68.87 | $29.07 \pm 0.48$ | 2.87 | $21.55 \pm 1.22$ | 19.49 |
| N-MC OI | $5.43 \pm 1.01$ | 4.79 | $-7.71 \pm 0.91$ | 7.05 | $23.91 \pm 1.63$ | 4.68 | $46.21 \pm 1.94$ | 5.55 |
| N-MC IPW | $5.23 \pm 0.93$ | 4.97 | $-7.43 \pm 0.93$ | 7.33 | $25.29 \pm 1.72$ | 3.47 | $46.23 \pm 1.93$ | 5.57 |
| N-OI | $4.36 \pm 2.05$ | 6.09 | $\mathbf{-15.90 \pm 1.14}$ | $\mathbf{1.65}$ | $31.21 \pm 1.68$ | 3.36 | $44.91 \pm 1.46$ | 4.17 |
| N-IPW | $\mathbf{8.83 \pm 0.36}$ | $\mathbf{1.33}$ | $-12.21 \pm 1.09$ | 2.72 | $\mathbf{27.90 \pm 0.99}$ | $\mathbf{1.06}$ | $\mathbf{42.60 \pm 2.02}$ | $\mathbf{2.58}$ |
| Ground Truth | $\mathbf{10.11}$ [NCT03987919, 18] | | $\mathbf{-14.7}$ [NCT03191396, 8] | | $\mathbf{28.3}$ [NCT03828539, 37] | | $\mathbf{41.00}$ [NCT02191579, 39] | |

