# OpenReview forum: "End-To-End Causal Effect Estimation from Unstructured Natural Language Data"
_NeurIPS.cc/2024/Conference — NeurIPS 2024 poster_

### Official Review · Reviewer_p9SY · 2024-07-08

**Soundness:** 3
**Presentation:** 1
**Contribution:** 3
**Rating:** 4
**Confidence:** 4

**Summary:**

This work aims to the causal effect estimation from unstructured observational text data, proposing a pipeline based on LLM to extract the treatment, covariate and outcome from the unstructured data by LLM. It's a interesting exploration for causla inference and the pipeline process of the inference method is quite standard.

**Strengths:**

(1) The problem solution for treatment effect estiamtion is quite novel.

(2) The proposed LLM pipeline can address the limitation of the current effect estimation method, i.e., can not deal with the unstructured data.

**Weaknesses:**

(1) some details are missing, leading to the poor readability.

(2) Some equations are not correct (based on my understanding).

(3) The presnetation of the whole methodology is not good enough, alought I am not an expert of LLM, I know how LLM works. However, i can not imagine how the proposed pipeline work and how it estimate ATE very clearly.

(4) The evaluation part is kind of weak, given that baselines are too limited and simple.

**Questions:**

1.there are some claims that are not well expalined, leading to confuse people. For exampel, "value may be recouped from outcomes that would otherwise be lost" this claim sounds good but i don't what does it mean here.

2.the overall idea of this work to utilize LLM to explore the causal effect estimation from unstructured data is good, i like it. But the writing can not match the good idea. For example, line 54-59, i have no idea about what the authors try to express, The transition from sentence to sentence is too stiff

3.The binary and discrete assumption about the outcome Y and covariates X is kind of too strict. Typically, one can assume the teatment assignment is binary but outcome and covariates are continous. It's not clear why this work adopt such assumption. if so, the applying scale of the proposed method would be limited. also, the assumption 4 is also too strong.

4.in the  outcome predictor eq. (9), why it is P(Y=1|R_i,X_i,T_i) instead of P(Y=t|R_i,X_i,T_i)? I don't understand. When t=0, the outcome predictor still ouput the expectation over P(Y=1|xxx)?

5.the description how to extrct the covariates and their corresponding values is not clear.  How to determine the feature values? For each report, the set of covarivates is the same, right? Can you present a specific example? what does each feature value mean in the real-world? The readability of the method section is poor.

6.In the evaluation table 1, the "IPW (Structured) " is a baseline or ground truth? there is no any discussion in the main body about this baseline and i am so confused about the discussion below because of it, given that the "IPW (Structured) " performs the best.

7.how does the ground truth of ate comes from? Can you clarify this point?

by the way, i like the idea of this work, but the presentation of this paper is really poor. thus if the above questions can be addressed well, i will consider to increase the score.

**Limitations:**

authors have adequately addressed the limitations.

---

> ### Author Rebuttal · Authors · 2024-08-07
>
> Thank you for your review and questions! Below, we will clarify all the questions in the review.
>
> > (W1) some details are missing, leading to the poor readability.
>
> Could the reviewer point to specific details that they found unclear? We are more than happy to provide clarification.
>
> > (W2) Some equations are not correct (based on my understanding).
>
> Similar to above, if the reviewer can point out particular equations that they found to be incorrect, we are happy to clarify them.
>
> > (W3) The evaluation part is kind of weak, given that baselines are too limited and simple.
>
> - Since we propose a novel setting of constructing causal effect estimators from unstructured text data, there aren’t a lot of existing baselines. We constructed fair comparisons and included ablation studies for our experiments. In synthetic settings, we were also able to train baselines that require additional structured data (which NATURAL does not assume access to) and presented those results in Table 1. In real-world settings, we further teased apart the effect of different choices we made in the method implementation, as shown in Figure 4.
> - If the reviewer has specific suggestions for baselines that are relevant to our setting, we would be happy to include them.
>
> > (Q1) there are some claims that are not well expalined, leading to confuse people. For exampel, "value may be recouped from outcomes that would otherwise be lost" this claim sounds good but i don't what does it mean here.
>
> We apologize for any confusion and expand on the referenced phrase here.
> - Sometimes there are outcomes that individuals might observe themselves, that are not captured in structured, tabulated observational data, but that are captured in unstructured text data on the internet.
> - Had these outcomes been tabulated and recorded, they could have contributed in valuable ways to decision-making.
> - NATURAL recovers the value of these outcomes for decision-making by extracting them from unstructured text data.
>
>
> We are happy to change this sentence to “value may be *recovered* from outcomes that would otherwise be lost” in the paper.
>
> > (Q2) the overall idea of this work to utilize LLM to explore the causal effect estimation from unstructured data is good, i like it. But the writing can not match the good idea. For example, line 54-59, i have no idea about what the authors try to express, The transition from sentence to sentence is too stiff
>
> The referred lines are intended to explain the different information NATURAL depends on. These are: (1) LLMs to approximate relevant conditional distributions, and (2) expert knowledge to ensure necessary causal assumptions are satisfied by the study design.
>
> > (Q3) The binary and discrete assumption about the outcome Y and covariates X is kind of too strict. Typically, one can assume the teatment assignment is binary but outcome and covariates are continous. It's not clear why this work adopt such assumption. if so, the applying scale of the proposed method would be limited.
>
> - Our NATURAL Full estimator requires discrete and finite variables since we sum over them to approximate a conditional expectation with conditional probabilities. If we choose to use samples instead of conditionals, NATURAL can further be extended to continuous variables.
> - While we have binary outcomes in our experiments, the method and implementations extend to non-binary discrete outcomes easily.
> - It is true that outcomes and covariates in the real world are often continuous. In order to keep NATURAL applicable in these settings, we discretize these variables. We will explain the exact discretization through a complete walkthrough of the Semaglutide vs. Tirzepatide dataset in the appendix.
>
> > also, the assumption 4 is also too strong.
>
> Assumption 4 is weak or strong depending on how much information is in the reports. Our data filtration pipeline aims to maximize the information in the reports so that Assumption 4 may be matched. We discuss this in more detail in Section 3 of the paper as well as our response to Q1 from reviewer 7S1Z.
>
> > (Q4) in the outcome predictor eq. (9), why it is P(Y=1|R_i,X_i,T_i) instead of P(Y=t|R_i,X_i,T_i)? I don't understand. When t=0, the outcome predictor still ouput the expectation over P(Y=1|xxx)?
>
> - We don’t believe there is a mistake in eq. (9). It is a Monte Carlo approximation of the first term in eq. (2). Eq. (2), in turn, is the standard form of the Outcome Imputation estimator (see Ding (2023)).
> - To clarify our notation, $Y$ is a random variable denoting outcomes and $T$ is a random variable denoting treatments. Therefore, ${Y=t}$ is not an event that any of our estimators would consider.
>
> [1] Ding, P. (2024). A first course in causal inference. CRC Press.
>
> > (Q5) the description how to extrct the covariates and their corresponding values is not clear. How to determine the feature values? For each report, the set of covarivates is the same, right? Can you present a specific example? what does each feature value mean in the real-world? The readability of the method section is poor.
>
> - While the definition of the covariate variables and their descriptions are the same for reports in a given dataset, the values these variables take will vary for each report. For instance, if a covariate of interest is “age”, its value in the real world may be a different number for different reports.
> - We will include more details and exact descriptions of each variable for each dataset in the appendix of our final paper.

---

> ### Author Response · Authors · 2024-08-07
> **Rebuttal by Authors (continued)**
>
> > (Q6) In the evaluation table 1, the "IPW (Structured) " is a baseline or ground truth? there is no any discussion in the main body about this baseline and i am so confused about the discussion below because of it, given that the "IPW (Structured) " performs the best.
>
> Thank you for pointing this out! We will clarify this result in the experiments section of the main paper.
> - IPW (Structured) corresponds to an oracle estimator that has access to the true structured treatments, outcomes and covariates, which can be plugged into the IPW estimator to estimate ATE.
> - Hence, it is not directly comparable to methods that do not have access to structured data, like NATURAL. We expect it to provide an upper bound on the performance, as verified in the results of Table 1.
>
> > (Q7) how does the ground truth of ate comes from? Can you clarify this point?
>
> As mentioned in lines 68-69 and again in lines 281-284, for every dataset we consider, there exists a randomized controlled trial which provides a gold standard estimate of the ATE. We treat this ATE value as “ground truth”.
>
> We thank the reviewer for bringing up questions about the parts of our paper they found confusing. We believe the clarifications above improve the readability of our paper. We hope our responses address all their concerns and that they will consider increasing their score accordingly. We are more than happy to provide any further clarifications during the discussion period.

---

> > ### Comment · Area_Chair_jMNr · 2024-08-13
> >
> > Dear Reviewer p9SY,
> >
> > Thanks for your review! The discussion is ending soon, and it'd be greatly appreciated if you could acknowledge the author's rebuttal and update your review if necessary.
> >
> > Thank you!
> > AC

---

### Official Review · Reviewer_oSFK · 2024-07-12

**Soundness:** 4
**Presentation:** 4
**Contribution:** 3
**Rating:** 8
**Confidence:** 4

**Summary:**

The paper introduces a family of causal effect estimators named NATURAL, designed to use LLMs for mining causal effect estimates from observational text data. The authors address the challenge of automating data curation and using LLMs to impute missing information, presenting a method that conditions on structured variables to assist in computation of causal effect estimators. The authors develop six datasets (two synthetic and four real) to evaluate their method, and demonstrate that NATURAL estimators produce causal effect estimates within close range of ground truth from randomized trials.

**Strengths:**

1. Clear and thoughtful use of LLMs to estimate causal effects from text data. The authors present a cool methodology, properly formalized, for using LLMs to extract and impute structured variables from observational text data.

2. Development and evaluation of six datasets, including real-world clinical trial data.

3. Strong results showing causal effect estimates are within a close range of ground truth values.

**Weaknesses:**

1. The original claims made in the abstract+intro seems to be too grandiose, but in fact it is about the use of LLMs to classify variables of interest from the text. I would change the framing of the paper a bit to better reflect that. It would make the claims of the paper less objectionable and would better demonstrate the usefulness of NATURAL.

2. There is a vast literature in the social sciences on recovering interpretable variables directly from text for estimating causal effects from observational data. Most of these approaches rely on probabilistic approaches like topic models, but recently there’s growing interest in doing so automatically with LLMs. The authors should better address this literature, and compare their approach to relevant methods if possible.

3. The approach is really cool, but like any ML-pipeline approach it surely introduces errors at each step of the process (see for example Egami et al @ NeurIPS 2023). Each error will then bias the causal effect estimation. From the results of the paper I’m convinced empirically your approach works well, but would I be able to understand whether it produces accurate estimates on a new dataset? A deeper analysis and perhaps a way to propagate error/uncertainty to the estimation would be very helpful here.

**Questions:**

1. Authors explain that as part of their process, they “filter for reports that are likely to conform to the experimental design”. Could this introduce bias?

2. All of the “classical strategies” used assume no-hidden confounding. Could you imagine how someone would use an LLM with a more complex identification strategy (say IV, or RDD)? You claim at the end of the intro that you can anticipate someone doing so, I wonder if you can elaborate.

**Limitations:**

The authors have addressed the limitations of their work, acknowledging the dependency on the quality and calibration of LLMs and the potential inaccuracies in approximating the required conditional distributions. They emphasize that NATURAL estimators should not be used for high-stakes decision-making without experimental validation and stress the need for domain expert involvement to ensure causal assumptions are met.

---

> ### Author Rebuttal · Authors · 2024-08-07
>
> Thank you for such a positive and encouraging review! Below, we address all the questions raised in your review.
>
> > (W1) The original claims made in the abstract+intro seems to be too grandiose, but in fact it is about the use of LLMs to classify variables of interest from the text. I would change the framing of the paper a bit to better reflect that. It would make the claims of the paper less objectionable and would better demonstrate the usefulness of NATURAL.
>
> Thank you for this feedback! The two take-homes that we tried to emphasize in the abstract and introduction were (1) unstructured text data can be used as a *sole* source of rich causal effect information and (2) LLMs can be used to extract this information. We also believe it is important to clearly distinguish our data access setting from the classical settings of causal inference (see response to W2 below). Still, we’re more than happy to make edits to get the tone right, if the reviewer can make some specific suggestions or point out specific sentences that are objectionable.
>
> > (W2) There is a vast literature in the social sciences on recovering interpretable variables directly from text for estimating causal effects from observational data. Most of these approaches rely on probabilistic approaches like topic models, but recently there’s growing interest in doing so automatically with LLMs. The authors should better address this literature, and compare their approach to relevant methods if possible.
>
> - Thank you for pointing out this section of the literature! We would greatly appreciate any specific references from the reviewer. To reiterate the setting considered for NATURAL, we look at data sources that *only* contain natural language. This contrasts our setting from works like Falavarjani et al. (2017), which rely on a combination of text and numerical or tabular data. The ability to use only text data is important and motivated by settings where tabular data is unavailable or incomplete, e.g. neglected diseases, unrecorded abortions in certain countries, or illicit drug use. To our knowledge, such a setting with text as the sole source of information has not been considered yet.
> - Another example of related work that we found is Ahrens et al. (2021), which estimates latent topics or variables from text, and is relevant to the broad problem we consider. NATURAL is distinct in two ways. First, similar to above, Ahrens et al. rely on tabulated numerical outcomes as well as text data, while NATURAL operates using only text data. Second, NATURAL relies on domain expertise to provide a study design and in particular, define the treatment, outcome and covariate variables to be extracted from text, whereas Ahrens et al. discover these variables in the text. Extending NATURAL to remove dependence on domain expertise may be interesting future work, but would require further assumptions on LLM capabilities.
> - We will include the above citations and discussion in the final version of our paper.
>
> [1] Falavarjani, S. M., Hosseini, H., Noorian, Z., & Bagheri, E. (2017, May). Estimating the effect of exercising on users’ online behavior. In Proceedings of the International AAAI Conference on Web and Social Media (Vol. 11, No. 1, pp. 734-738).
>
> [2] Ahrens, M., Ashwin, J., Calliess, J. P., & Nguyen, V. (2021). Bayesian topic regression for causal inference. arXiv preprint arXiv:2109.05317.
>
> > (W3) The approach is really cool, but like any ML-pipeline approach it surely introduces errors at each step of the process (see for example Egami et al @ NeurIPS 2023). Each error will then bias the causal effect estimation. From the results of the paper I’m convinced empirically your approach works well, but would I be able to understand whether it produces accurate estimates on a new dataset? A deeper analysis and perhaps a way to propagate error/uncertainty to the estimation would be very helpful here.
>
> Your intuition is correct in that different errors can arise due to the assumptions required by NATURAL, some of which may not be formally testable yet. While this is important future work that we are actively pursuing, there are some approaches we can already use to mitigate and/or understand such errors.
> - We try to minimize errors in the steps using LLMs via prompt tuning on a handful of examples. We will describe this tuning in full detail in the appendix via a worked example that goes through all the steps of NATURAL for the Semaglutide vs. Tirzepatide dataset.
> - For methods that involve propensity score estimation, we use the balancing score property of propensity scores as a sanity-check. Figure 5 of our paper confirms that propensity scores estimated by NATURAL do indeed balance covariates across treatment groups.
> - We will also add a sensitivity analysis to gain more insight into the satisfaction of our Assumption 1 about Strong Ignorability or Unconfoundedness (see response to Q1 of Reviewer oSFK and attached PDF).
>
> > (Q1) Authors explain that as part of their process, they “filter for reports that are likely to conform to the experimental design”. Could this introduce bias?
>
> - The purpose of filtering for reports that conform to the experiment design is to estimate local ATEs within a specific population. This also makes our estimators comparable to the treatment effect from real-world RCTs that enforce inclusion criteria for participants. In practice, we use LLMs for this step, which can definitely introduce bias and is a limitation of our work, as discussed in Section 5.
> - In general, we do not expect one estimated ATE to transport to a different experiment design. Given another experiment design, one would have to execute the NATURAL pipeline with that design and estimate a new ATE. One advantage of NATURAL is that doing this for different designs is feasible in a time- and cost-effective manner.

---

> ### Author Response · Authors · 2024-08-07
> **Rebuttal by Authors (continued)**
>
> > (Q2) All of the “classical strategies” used assume no-hidden confounding. Could you imagine how someone would use an LLM with a more complex identification strategy (say IV, or RDD)? You claim at the end of the intro that you can anticipate someone doing so, I wonder if you can elaborate.
>
> This is a great question! The general strategy behind deriving NATURAL estimators from classical strategies is as follows:
> - Identify all (structured) variables required to compute the classical estimator. In the case of the Instrumental Variables (IV) approach, this includes variables representing the instrument $Z$, in addition to the $(T, Y)$ that our methods consider.
> - Next, collect natural language reports that contain such information. For instance, electronic health records or clinical notes at hospitals may contain information relevant to the IV setting.
> - Extract the required conditional distributions from an LLM using these natural language reports.
> - Finally, marginalize out $(T, Y, Z)$ and average over reports to compute an ATE. For example, IV estimators are used to measure ATE conditional on compliance, which is a ratio of two treatment effects (see chapter 21 of Ding (2024)) and can be estimated consistently given the reports.
>
> [1] Ding, P. (2024). A first course in causal inference. CRC Press.
>
> Again, thank you for your very thoughtful review! We hope we have addressed all your questions, but please let us know if we can clarify anything further.

---

> > ### Comment · Reviewer_oSFK · 2024-08-12
> >
> > thank you addressing my comments!
> > i remain supportive of this paper's acceptance.

---

### Official Review · Reviewer_7S1Z · 2024-07-13

**Soundness:** 3
**Presentation:** 3
**Contribution:** 3
**Rating:** 7
**Confidence:** 3

**Summary:**

Estimating causal effects is costly and time-consuming. The authors propose  to use large language models (LLMs) to mine unstructured text data for causal effect estimation. This paper introduces NATURAL, a family of causal effect estimators using LLMs to process unstructured text data. This seems to be a good application of LLMs for the task of causality.

**Strengths:**

-  Determine the effects of treatments can be expensive and time-consuming so the paper introduces a novel task of effect estimation using unstructured natural language data.

- This method is adaptable to various domains where textual data is abundant.

- This seems to be a more practical application of LLM for the task of causality.

**Weaknesses:**

- LLMs might inherit biases present in their training data or hallucinate.

 - Social media users might not reflect the demographics or behaviors of the broader population.

- Finetuning the model to one domain may not lead to good generalisation in others, during fine-tuning does the LLM learn inductive biases regarding the "topic" or task?

**Questions:**

-  How do the authors make sure that all of the confounders are accounted ? I can imagine there to be many confounders mentioned in the text.

- There could be a bias in social media text, I thought people would report negative cases more often than positive, did you observe any such biases?

**Limitations:**

Mentioned

---

> ### Author Rebuttal · Authors · 2024-08-07
>
> Thank you for your helpful review! Here, we will clarify and address the points brought up in your review.
>
> > (W1) LLMs might inherit biases present in their training data or hallucinate.
>
> As described in Section 5, this is an important limitation in the use of LLMs, in general as well as in the case of NATURAL. In our experiments, we mitigate errors or hallucinations via prompt tuning. We believe NATURAL will benefit from progress in the field of research with LLMs, but the specific challenge is outside the scope of our work.
>
> > (W2) Social media users might not reflect the demographics or behaviors of the broader population.
>
> - It is true that there exists selection bias in the data on social media forums and this is an important limitation that we’re hoping to explore in future work. We discuss this and other biases in Appendix E of the submission and plan to expand Section 5 with this discussion in the final version of our paper.
> - Related to this is the treatment of inclusion criteria that define the population over which we are interested in estimating ATEs. In Appendix G of the paper, we made certain structural assumptions that allowed us to impute unobserved covariates restricted to the inclusion criteria. An alternate approach that we also discussed in Appendix G is to weight the estimated potential outcomes for each datapoint by the relative likelihood that they meet the inclusion criteria of the experiment given the report: $\frac{P(X \in I | R)}{P(X \in I)}$ where $I$ defines the inclusion criteria. We revisited this alternate approach during the rebuttal period and explored using the LLM to estimate $P(X_i \in I | R_i)$ and renormalizing to estimate  $\frac{P(X_i \in I | R_i)}{P(X_i \in I)}$. These weights can be estimated with an LLM similar to other conditional distributions described in the paper and involves averaging over reports for the denominator.
> - We show these new results in Table 2 of the attached PDF, which are close to our original results. We believe this alternate weighting approach is a simpler, more intuitive way to understand the treatment of inclusion criteria that also removes the need for additional assumptions. We will include both original and new results in the final draft.
>
> > (W3) Finetuning the model to one domain may not lead to good generalisation in others, during fine-tuning does the LLM learn inductive biases regarding the "topic" or task?
>
> To clarify our use of LLMs, none of our methods require any fine-tuning. Instead we use LLMs purely for inference. We will clarify this point in Section 4 of the paper.
>
> > (Q1) How do the authors make sure that all of the confounders are accounted ? I can imagine there to be many confounders mentioned in the text.
>
> - We rely on domain expertise to define the confounders for each setting such that necessary causal assumptions are satisfied. In our case, we used baseline characteristics from real-world RCT specifications which are designed by experts. Together with Assumption 4, which guarantees access to the conditional distribution $P(T = t, Y = y, X = x|R = r)$, these assumptions ensure that our estimators are consistent in theory.
> - Intuitively, we break this problem into two parts: (1) defining the confounder set $X$ such that Assumption 1 (Strong Ignorability or Unconfoundedness) is satisfied, and (2) correctly sampling these $X$ from $R$, which is guaranteed by Assumption 4 (in theory) and implemented with the help of LLMs (in practice).
> - Still, we think you bring up a really important point. So, we conducted a sensitivity analysis for the Semaglutide vs. Tirzepatide dataset. Specifically, we follow the sensitivity analysis strategy of Lu and Ding (2023). Briefly, the idea is to introduce sensitivity parameters ($\varepsilon_0(X) = \frac{\mathbb{E}[Y(0) | T=1,X]}{\mathbb{E}[Y(0) | T=0,X]}$ and $\varepsilon_1(X) = \frac{\mathbb{E}[Y(1) | T=1,X]}{\mathbb{E}[Y(1) | T=0,X]}$) that quantify the degree of unobserved confoundedness. In our case, sensitivity parameters are the density ratio between the likelihood of each potential outcome in the treated vs. untreated group. The ATE estimate is non-increasing in the sensitivity parameters. So, for positive ATEs, we are looking for the largest sensitivity parameters that maintain the positivity of the ATE, which tells us the degree of unobserved confoundedness that an estimator is robust to. Table 1 of the attached PDF shows that the estimated ATE changes from positive to negative at large values of sensitivity parameters, which means that it is robust to large degrees of unobserved confoundedness.
> - We further investigate the importance of each confounder as suggested by Section 4 of Lu and Ding (2023), by dropping that covariate as if it is an unobserved confounder and measuring the corresponding sensitivity parameters with the remaining covariates. Figure 1 of the attached PDF follows Figure 1 of Lu and Ding (2023) and shows that in the worst-case (over all possible values of the remaining covariates), our estimator is sensitive to the covariate set considered and each covariate is important to ATE estimation.
> - We believe this sensitivity analysis significantly improves our paper and will include it in the final version of our paper. Thanks for your feedback!
>
> [1] Lu, S., & Ding, P. (2023). Flexible sensitivity analysis for causal inference in observational studies subject to unmeasured confounding. arXiv preprint arXiv:2305.17643.

---

> ### Author Response · Authors · 2024-08-07
> **Rebuttal by Authors (continued)**
>
> > (Q2) There could be a bias in social media text, I thought people would report negative cases more often than positive, did you observe any such biases?
> - Since this relates to the concern about selection bias above, please see our response to W2 above.
> - Since our experiments compare similar treatments, e.g., comparable availability, we believe the probability of a user reporting their experience is approximately equal in both. Our empirical results suggest low bias, relative to estimates from a real-world RCT.
>
> Thank you for the great suggestions and questions! We believe our paper is significantly improved by the edits and clarifications above. We would be happy to clarify anything further in the discussion period.

---

> > ### Comment · Reviewer_7S1Z · 2024-08-10
> > **Thank you for rebuttal**
> >
> > I would like to thank the reviewer for replying to my concerns regarding biases and confounders. I have increased my score.
> >
> > Best regards,

---

### Official Review · Reviewer_oMqj · 2024-07-14

**Soundness:** 2
**Presentation:** 3
**Contribution:** 3
**Rating:** 7
**Confidence:** 3

**Summary:**

The paper seeks to use LLMs for a workflow that extracts structured variables
from free text, filters the dataset, and then computes a causal estimate using
the imputed variables. The paper applies this methodology to two synthetic and
four real-world datasets and shows that it produces estimates that are
comparable to those from known ground truth or RCT estimates.

**Strengths:**

This is a very ambitious paper that pulls together a complicated workflow for
estimating causal effects from real-world natural language datasets. Just
showing that this can be done is a significant contribution.

The paper is clearly written and the main points are easy to follow.

While I am somewhat skeptical of any practical applicability of this work, the
paper is reasonable conservative in highlighting its limitations.

**Weaknesses:**

The paper at almost every turn relies on prompting an LLM and implicitly
assuming that the model returns data from a desired distribution. In general,
this will introduce measurement error which could systematically bias the
overall method's estimates. See for example [1] and [2] below.

The synthetic data evaluation is fairly limited. It would be nice to at least
understand how performance varies as sample size increases from a few hundred
examples to many thousands (especially as the IPW oracle baselines are not
particularly good). This would also allow you to stress-test the different
assumptions being made. If you have synthetic data in which reports for certain
patients are systematically truncated or censored, or where different groups
write reports in different languages or styles, how does that affect the
different steps (i-vi)?

[1] Ashwin, Julian, Aditya Chhabra, and Vijayendra Rao. "Using Large Language
Models for Qualitative Analysis can Introduce Serious Bias." Development
Research (2023).

[2] Egami, Naoki, et al. "Using imperfect surrogates for downstream inference:
Design-based supervised learning for social science applications of large
language models." Advances in Neural Information Processing Systems 36
(2024).

**Questions:**

Line 267-268: What is the point of sampling a persona from Big Five personality
traits? How does this simulate realism?

Line 147-149 says "if all reports are the constant, empty string ... we have
full access to the true observational joint density." Am I correct that this is
a necessary assumption for the overall method to work in theory, but the actual
implementation (steps (v) and (vi) in lines 206-220) would not work if all
reports are the constant, empty string? This may be obvious, but it could be
worth clarifying that when you say "we cannot formally guarantee that [our
method] satisfies Assumptions 3 and 4" you mean the form of Assumption 4 that
does not involve all reports being empty strings.

In Tables 1 and 2, which LLMs are used? LLAMA2-70B? Overall, it would be
helpful to more clearly label which LLMs are used when. In Line 204, the paper
says that the GPT-4 API sufficed for filtering; was GPT-4 used anywhere else?
How was the decision made of which LLM to use where?

It might be helpful to have a figure or appendix that walks through every step
(the paper does this for many steps, just not all in one place) in which an
LLM is queried and discusses what assumptions are being made.

**Limitations:**

Overall, the paper does a good job of not overclaiming. However, see weakness 1
above. I would also suggest the authors discuss the challenges of using RCT
estimates to validate observational studies -- there are many reasons why those
estimates might not align (selection bias, dropout, etc).

---

> ### Author Rebuttal · Authors · 2024-08-07
>
> Thank you for your detailed and positive feedback! Below, we address the questions in your review and describe corresponding changes to improve our paper.
>
> > (W1) The paper at almost every turn relies on prompting an LLM and implicitly assuming that the model returns data from a desired distribution. In general, this will introduce measurement error which could systematically bias the overall method's estimates.
>
> We are able to measure this error in synthetic settings, which we quantify as KL divergence between the true and predicted distributions and visualize in Figure 3. With increasing amounts of data, this KL divergence reduces, as does the ATE error.
> Nevertheless, we agree that measurement error due to an LLM’s output distribution is a potential limitation of our work, as described in Section 5. We discussed selection and other possible biases in more detail in Appendix E of our submission due to space constraints. We will include this discussion in the main paper in the final version.
>
> > (W2) The synthetic data evaluation is fairly limited. It would be nice to at least understand how performance varies as sample size increases from a few hundred examples to many thousands (especially as the IPW oracle baselines are not particularly good). This would also allow you to stress-test the different assumptions being made. If you have synthetic data in which reports for certain patients are systematically truncated or censored, or where different groups write reports in different languages or styles, how does that affect the different steps (i-vi)?
>
> - We would like to clarify that the oracle estimates in Table 1 are denoted by “IPW (Structured)”. We will update the discussion of results in Section 6 to clarify this.
> - We constructed synthetic datasets of 2000 reports each and found these IPW (Structured) oracle estimates to be very close to the ground truth ATE, as sample size was increased to 1024 or higher. If the reviewer thinks it is important, we are happy to generate more synthetic data and show performance at higher sample sizes.
> - Simulating systematic biases in synthetic settings is a great suggestion and important future work! For example, one could simulate reporting bias by subsampling the data based on a subset of the covariates. We ran this during the rebuttal period for the Hillstrom dataset using two covariates (channel and zip code). Indeed this results in a NATURAL IPW ATE of $-3.49$ which has a large error of $9.58$. We will include this result in Table 1 as an example of NATURAL’s current limitation in handling reporting bias, which we hope to address in future work.
> - We are also open to further discussion on how to investigate data that is biased in other ways, such as being “truncated”, if the reviewer has concrete suggestions.
>
> > (Q1) What is the point of sampling a persona from Big Five personality traits? How does this simulate realism?
>
> The Big Five personality traits (Openness, Conscientiousness, Extraversion, Agreeableness, and Neuroticism) helped simulate diversity in writing style, tone and report lengths, and hence made the synthetic data more realistic.
> We observed the generated reports were diverse in their verbosity, style and tone, as one might expect in real-world online forums.
>
> > (Q2) Am I correct that this is a necessary assumption for the overall method to work in theory, but the actual implementation (steps (v) and (vi) in lines 206-220) would not work if all reports are the constant, empty string?
>
> Your understanding is correct! If the reports are all empty strings, then Assumption 4 requires that the LLM is able to simulate trial outcomes unconditionally (without any data). This would be a very strong assumption that is likely to be violated. We will clarify this in Section 3 of the paper. In practice, we filter our data to look for the most informative reports such that they contain some information about $(X,T,Y)$, as described in Section 4.
>
> > (Q3) In Tables 1 and 2, which LLMs are used? LLAMA2-70B? Overall, it would be helpful to more clearly label which LLMs are used when. In Line 204, the paper says that the GPT-4 API sufficed for filtering; was GPT-4 used anywhere else?
>
> Thank you for pointing this out! We used GPT-3.5 Turbo for filtering steps, GPT-4 Turbo for sampling and LLAMA2-70B for computing conditional probabilities, for all datasets. We will explicitly state these details in Section 6.
>
> > (Q4) How was the decision made of which LLM to use where?
>
> - We tried different LLMs and chose the best performing ones available at the time, by examining a handful of examples at different points in the pipeline. We also took into account the costs associated with these models.
> - The filtering steps were relatively easy tasks, but were executed on larger amounts of data; hence we opted for the cheaper GPT-3.5 Turbo model. The sampling steps were tasks involving more reasoning and were executed on smaller datasets that had already been filtered; hence we opted for the best performing model at the time: GPT-4 Turbo. The LLM conditional probabilities required access to log-probabilities, and we opted for LLAMA2-70B as it was a top open-source model providing this functionality. An ablation on the scale of the model used for this step is included in Figure 4 (right).
>
> > (Q5) It might be helpful to have a figure or appendix that walks through every step (the paper does this for many steps, just not all in one place) in which an LLM is queried and discusses what assumptions are being made.
>
> This is a great suggestion; thank you! We will add a section in the appendix that walks through a complete worked example of executing our pipeline with the Semaglutide vs. Tirzepatide dataset.

---

> ### Author Response · Authors · 2024-08-07
> **Rebuttal by Author (continued)**
>
> > I would also suggest the authors discuss the challenges of using RCT estimates to validate observational studies -- there are many reasons why those estimates might not align (selection bias, dropout, etc).
>
>
> Thank you for this suggestion! Since these are important considerations for future work, we will expand Section 5 of the paper to elaborate on the limitations of NATURAL.
>
> Thank you for all your suggestions, which we believe will greatly improve our paper! Please let us know if there are any further questions we can address.

---

> > ### Comment · Reviewer_oMqj · 2024-08-11
> >
> > I appreciate the authors’ careful response. The planned additions and clarifications to the paper will strengthen it.
> >
> > > We would like to clarify that the oracle estimates in Table 1 are denoted by “IPW (Structured)” […] oracle estimates to be very close to the ground truth ATE
> >
> > This was clear to me when writing my original review, but from my perspective Table 1 shows “closer” but not “very close” estimates. Given the many limitations of synthetic data (in general and specifically in the comparison between Hillstrom and Retail Hero compared to the domain to which you apply this method), I think it would be very useful to show that the IPW (structured) oracle converges to an arbitrarily low RMSE as sample size increases. That then provides a better comparison for how well the NATURAL methods are empirically performing compared to a method that theoretically should and practically does achieve unbiased estimation.
> >
> > Two follow-up questions that I didn’t catch in my original reading:
> > - What explains the mismatch between the low RMSE in Figure 3 (close to $10^{-2}$ on both synthetic datasets) and the N-FULL RMSEs in Table 1?
> > - Can you clarify the specifics of the ATE (%) versus RMSE metrics in Table 1? How are the standard errors being calculated? Are there any insights from the mismatches between the two metrics (e.g., N-MC OI on Hillstrom and N-OI on RetailHero have closer average ATEs to the ground truth but worse RMSE)?
> >
> > > We are also open to further discussion on how to investigate data that is biased in other ways, such as being “truncated”, if the reviewer has concrete suggestions.
> >
> > Perhaps the easiest way to do this would be to have a “truncation model” that’s conditional on covariates (e.g., $\mathbf{P}(\text{Truncation}) = \text{logit}(w_0 + w_c\cdot \text{Channel} + w_z\cdot\text{ZipCode})$ or similar) where if you sample Truncation=1, you truncate the corresponding example’s report to contain only the first $k$ tokens. The real-world analogue I think is likely applicable is that there may be systematic biases in terms of the availability or verbosity of certain groups’ reports. This is a somewhat heavy-handed way to imitate that, but it should be easy to implement.
> >
> > > helped simulate diversity in writing style, tone and report lengths, and hence made the synthetic data more realistic.
> >
> > Thanks for the clarification. I was interpreting this as a claim that the Big Five traits were specifically relevant to these synthetic datasets, rather than this is a general way to introduce diversity in generation.

---

> > > ### Author Response · Authors · 2024-08-12
> > >
> > > Thank you for your comments on our rebuttal!
> > >
> > > > I think it would be very useful to show that the IPW (structured) oracle converges to an arbitrarily low RMSE as sample size increases.
> > >
> > > Thanks for clarifying your point further! We can demonstrate the convergence of the IPW (Structured) oracle to ground truth on a larger number of datapoints. Using all 2000 data points of the observational Hillstrom dataset, the oracle gives an ATE estimate of 6.08 (on a scale from -100 to 100), as compared to the ground truth of 6.09. To test and compare to NATURAL methods on even larger samples of data, we would need to generate more synthetic reports.
> > >
> > > > What explains the mismatch between the low RMSE in Figure 3 (close to $10^{-2}$ on both synthetic datasets) and the N-FULL RMSEs in Table 1?
> > >
> > > Thank you for catching this - the apparent mismatch is due to the scale on which the results (ATE and RMSE) are reported. The tables use a scale from -100 to 100, while the plots use -1 to 1. To maintain consistency, we will update the plots to use the same scale as the tables in the final version of our paper.
> > >
> > > > Can you clarify the specifics of the ATE (%) versus RMSE metrics in Table 1? How are the standard errors being calculated? Are there any insights from the mismatches between the two metrics (e.g., N-MC OI on Hillstrom and N-OI on RetailHero have closer average ATEs to the ground truth but worse RMSE)?
> > >
> > > The ATE (%), standard error and RMSE are all calculated over 10 runs, sampling 80% of the data points without replacement in each one. The difference in performance based on mean ATE vs RMSE can be explained by the standard deviation of the estimates. For instance, N-MC OI on Hillstrom estimates average ATE closer to the ground truth, but has higher variance across runs.
> > >
> > > > Perhaps the easiest way to do this would be to have a “truncation model” that’s conditional on covariates (e.g., $\mathbf{P}(\text{Truncation}) = \text{logit}(w_0 + w_c\cdot \text{Channel} + w_z\cdot\text{ZipCode})$ or similar) where if you sample Truncation=1, you truncate the corresponding example’s report to contain only the first $k$ tokens. The real-world analogue I think is likely applicable is that there may be systematic biases in terms of the availability or verbosity of certain groups’ reports.
> > >
> > > Thanks for this suggestion! It makes complete sense to try and simulate varying verbosity as a function of individuals’ covariates. Since truncating to the first $k$ tokens for some $k$ will likely lead to incomplete words or sentences at the end of the report, these reports might not be very realistic data. Instead, we could use the “Openness” trait from the Big Five and set its probability to be a function of covariates (as in your example with channel and zip code). This could be a realistic way to vary verbosity because we found “Low Openness” to generate short and succinct reports, and vice versa. We are excited to explore these variations in future work!

---

### Author Rebuttal · Authors · 2024-08-07

We thank the reviewers for their time and the effort they took to provide valuable feedback!

Overall, all of the reviewers appreciated the significance and novelty of our work, with oMqj calling it “ambitious work” and with oSFK calling it a “cool” and “properly formalized” methodology. Most reviewers also appreciated the clarity of our presentation.

Reviewer feedback has helped us identify a few ways to improve the paper. Based on the reviewers’ suggestions, we would like to include the following edits. We expand on these points in more detailed reviewer feedback in the comments below.

1. **Complete walkthrough of NATURAL for Semaglutide vs. Tirzepatide (oMqj)**: We will add a section working through the entire NATURAL pipeline for the Semaglutide vs. Tirzepatide dataset, in the appendix. This will include a step-by-step explanation of how to implement our method as a function of the experiment design, as well as the strategies we used to minimize error in each step.

2. **Expanded section on limitations (oMqj and oSFK)**: We agree that it is very important to detail the limitations of NATURAL. We stated these in our paper and included an extended discussion of these in the appendix of our original submission, due to space constraints. We will consolidate all discussions addressing limitations in an expanded Section 5 of the final version of our paper.

3. **Sensitivity analysis to address the Strong Ignorability assumption (7S1Z)**: We have run an analysis of the sensitivity of our ATE estimates to the degree of unobserved confoundedness, following the strategy in Lu and Ding (2023). In the attached PDF, Table 1 shows that the direction (sign) of our estimates is robust to large changes in the degree of unobserved confoundedness. Further, Figure 1 shows that each covariate we accounted for is important to our ATE estimation, via a leave-one-covariate-out approach. We expand on the interpretation of these analyses in our comments to reviewer 7S1Z below.

4. **Intuitive way to treat inclusion criteria and partially address selection bias (oMqj and 7S1Z)**: Selection effects are important considerations for our work, which includes factors like inclusion criteria and reporting biases. Reporting bias is outside of the scope of our current paper, and we address it in the expanded limitations section (see appendix and above). When it comes to inclusion criteria, in Appendix G of the paper we made certain structural assumptions that allowed us to impute unobserved covariates restricted to the inclusion criteria. During the rebuttal period, we implemented an alternate approach, which is also described in Appendix G in the original paper, that allows us to remove these additional assumptions by estimating an additional probability with the LLM. We show these new results in Table 2 of the attached PDF, which are close to our original results and which we will include alongside the original results in the final draft. We describe this in more detail in our response to reviewer 7S1Z.

We look forward to engaging further during the discussion period to clarify anything else.

[1] Lu, S., & Ding, P. (2023). Flexible sensitivity analysis for causal inference in observational studies subject to unmeasured confounding. arXiv preprint arXiv:2305.17643.

---

### Decision · Program_Chairs · 2024-09-25

**Decision:**

Accept (poster)

**Comment:**

The paper develops an innovative, causal-aware way to estimate causal effects with the help of large language models. Overall, the reviewers including myself found the idea fresh and interesting. Here, I noted the term 'causal-aware' in the sense that the method does not have a theoretical grounding as many other papers in the causal inference track but offers a clever way together with strong empirics.

The reviewers had some comments on the presentation (which can hopefully addressed in the camera-ready version). The question regarding more baseline is a difficult one, given that the task is fairly novel, so that simply baselines are not available (which is often the case in causal inference).

Overall the reviewers find the contribution valuable and the authors have made a significant effort in addressing clarifying questions and concerns from reviewers. I do believe all additional experiments should be incorporated in the main paper. As a result, I think that the paper is ready to be accepted.